# The Identification of a Novel Nucleomodulin MbovP467 of *Mycoplasmopsis bovis* and Its Potential Contribution in Pathogenesis

**DOI:** 10.3390/cells13070604

**Published:** 2024-03-29

**Authors:** Abdul Raheem, Doukun Lu, Abdul Karim Khalid, Gang Zhao, Yingjie Fu, Yingyu Chen, Xi Chen, Changmin Hu, Jianguo Chen, Huanchun Chen, Aizhen Guo

**Affiliations:** 1National Key Laboratory of Agricultural Microbiology, Huazhong Agricultural University, Wuhan 430070, China; abdul@mail.hzau.edu.cn (A.R.); doukunlu@webmail.hzau.edu.cn (D.L.); drabdulkarim43@webmail.hzau.edu.cn (A.K.K.); fuyingjie@webmail.hzau.edu.cn (Y.F.); chenyingyu@mail.hzau.edu.cn (Y.C.); chenxi@mail.hzau.edu.cn (X.C.); hcm@mail.hzau.edu.cn (C.H.); chenjg@mail.hzau.edu.cn (J.C.); chenhch@mail.edu.cn (H.C.); 2Hubei Hongshan Laboratory, Huazhong Agricultural University, Wuhan 430070, China; 3College of Veterinary Medicine, Huazhong Agricultural University, Wuhan 430070, China; 4Hubei International Scientific and Technological Cooperation Base of Veterinary Epidemiology, Huazhong Agricultural University, Wuhan 430070, China; 5International Research Center for Animal Disease, Ministry of Science and Technology, Huazhong Agricultural University, Wuhan 430070, China; 6Key Laboratory of Ministry of Education for Conservation and Utilization of Special Biological Resources in the Western China, School of Life Sciences, Ningxia University, Yinchuan 750021, China; zhaogang@nxu.edu.cn

**Keywords:** *Mycoplasmopsis bovis*, MbovP467, nucleomodulin, pathogenesis, apoptosis, cytokines

## Abstract

*Mycoplasmopsis bovis* is a causative agent of crucial diseases in both dairy and beef cattle leading to substantial economic losses. However, limited control measures for *M. bovis*-related diseases exist due to a lack of understanding about the virulence factors of this pathogen, a common challenge in mycoplasma research. Consequently, this study aimed to characterize a novel nucleomodulin as a virulence-related factor of *M. bovis*. Employing bioinformatic tools, we initially predicted MbovP467 to be a secreted protein with a nuclear localization signal based on SignalP scores and the cNLS (Nuclear Localization Signal) Mapper, respectively. Subsequently, the MbovP467 gene was synthesized and cloned into a pEGFP plasmid with EGFP labeling to obtain a recombinant plasmid (rpEGFP-MbovP467) and then was also cloned in pET-30a with a consideration for an *Escherichia coli* codon bias and expressed and purified for the production of polyclonal antibodies against the recombinant MbovP467 protein. Confocal microscopy and a Western blotting assay confirmed the nuclear location of MbovP467 in bovine macrophages (BoMacs). RNA-seq data revealed 220 up-regulated and 20 down-regulated genes in the rpEGFP-MbovP467-treated BoMac group compared to the control group (pEGFP). A GO- and KEGG-enrichment analysis identified associations with inflammatory responses, G protein-coupled receptor signaling pathways, nuclear receptor activity, sequence-specific DNA binding, the regulation of cell proliferation, IL-8, apoptotic processes, cell growth and death, the TNF signaling pathway, the NF-κB signaling pathway, pathways in cancer, and protein families of signaling and cellular processes among the differentially expressed up-regulated mRNAs. Further experiments, investigating cell viability and the inflammatory response, demonstrated that MbovP467 reduces BoMac cell viability and induces the mRNA expression of IL-1β, IL-6, IL-8, TNF-α, and apoptosis in BoMac cells. Further, MbovP467 increased the promoter activity of TNF-α. In conclusion, this study identified a new nucleomodulin, MbovP467, for *M. bovis*, which might have an important role in *M. bovis* pathogenesis.

## 1. Introduction

Mycoplasmas, characterized by their minimal genomes and lack of cell walls, represent the smallest free-living prokaryotic pathogens [1]. Mycoplasmas, as obligate parasites, inhabit a diverse array of host species, including the majority of livestock animals and humans alike [2]. The limited understanding of mycoplasma pathogenesis has impeded the development of effective countermeasures against mycoplasmosis. Past investigations have identified virulence factors associated with adhesion [3], invasion [4,5], and inflammatory responses [6], suggesting their potential role in mycoplasma pathogenesis. Among the less-explored aspects, nucleomodulins, a category of proteins that can translocate into host cellular nuclei and manipulate various biological processes, have garnered attention. These nuclear-targeting proteins fall into distinct categories, namely nucleomodulins and cyclomodulins [7]. Nucleomodulins exhibit activities that modulate epigenetic processes [8], whereas cyclomodulins specifically disrupt the host cell cycle [9]. These cyclomodulins and nucleomodulins may govern the expression of host genes linked to immune responses, cellular proliferation, and apoptosis [10]. This regulation may be achieved through direct binding to the promoters of host genes. Additionally, these proteins interact with nuclear histones, thereby modifying epigenetic processes, potentially through mechanisms such as methylation, acetylation, and ubiquitination [11]. Furthermore, their interactions extend to epigenetic regulators, transcription or splicing factors, and signaling proteins. At present, the inventory of nucleomodulins encompasses at least 70 distinct proteins identified in bacterial pathogens infecting humans, animals, and plants [11].

Within the mycoplasma domain, our understanding of nucleomodulins remains limited. Some studies have been conducted to investigate the role of mycoplasma nucleomodulins in their pathogenesis as illuminated by Chernov et al.’s investigations. In their studies, DNA methyltransferases, originating from *M. hyorhinis*, were found to play a role in fostering tumorigenesis in cultured human cells. These methyltransferases selectively bind to GC- and GATC-specific motifs within the DNA of cellular target genes, thereby catalyzing DNA methylation processes [12]. Certain Mollicutes species, such as Aster Yellows phytoplasma, utilize nucleomodulin SAP11 to interact with and destabilize transcription factors responsible for regulating the development of plants [13]. Furthermore, investigations into *M. bovis* by Zhao et al. have shed light on the significant role of nucleomodulins in mycoplasma pathogenesis. Specifically, one nucleomodulin, MbovP475, of *M. bovis* was identified to induce apoptosis and reduce cell viability. This effect was mediated through the modulation of cell cycle central regulatory genes, namely αB-crystallin (CRYAB) and MCF2L2 genes [14,15]. In addition to this, some studies have investigated the secretome of certain mycoplasmas to enhance diagnostic tools and vaccines [16]. An analysis of *M. hyopneumoniae* and *M. flocculare* secretomes identified seven shared proteins. Recently, extracellular vesicles (EVs) from various mycoplasma species were studied. The smallest EVs were from *M. agalactiae* and *M. bovis*, while the largest were from *M. mycoides* subsp. *mycoides* Afadé [16]. Virulence factors were found in three species, including P37, OppA, LppB lipoproteins, Vpmas, MAG5040, Ef-Tu, Hsp70, and glucose permease, influencing various cellular processes and serving as EV markers [17]. However, their nucleomodulin status needs to be investigated. Therefore, this study was conducted to identify more mycoplasma nucleomodulins for *M. bovis*. As a result, MbovP467 was determined to be a novel mycoplasma nucleomodulin.

## 2. Materials and Methods

### 2.1. Ethics Statement

Our mouse experiment underwent an evaluation and approval by the local Ethical Committee of Huazhong Agricultural University, Wuhan, China (HZAUMO-2018-027). All experimental procedures and operations strictly followed the approved guidelines.

### 2.2. Growth of Cells and Bacterial Strains

The *M. bovis* strain HB0801 (GenBank accession no. NC_018077.1), isolated from a diseased beef cattle lung from Yingcheng city, China, [18], was cultured in pleuropneumonia-like organism (PPLO) media (BD Company, MD, Franklin Lakes, NJ, USA) containing 50% horse serum, using established protocols. The knockout strain T5.141, lacking MbovP467, was isolated from our laboratory’s transposon-mediated *M. bovis* mutant library [19]. The mutation occurred at nucleotide 334 of the Mbov_467 coding sequence, corresponding to nucleotide 543,809 of the *M. bovis* HB0801 genome. We cultured the mutant T5.141 in the same PPLO medium, supplementing it with either 100 μg/mL of gentamicin or 10 μg/mL of puromycin. *Escherichia coli* strain DH5α (TransGen, Beijing, China) was grown in a Luria–Bertani (LB) broth (Thermo Fisher, Waltham, MA, USA) at 37 °C with appropriate antibiotics as necessary. BoMac cells were grown in Dulbecco’s modified Eagle’s medium (DMEM), supplemented with 10% fetal bovine serum (FBS) and 1% antibiotics (penicillin-streptomycin). The plates were maintained in an incubator at 37 °C, 5% CO_2_, and 95% atmospheric air. The growth medium was refreshed every 2 days until the cells achieved appropriate confluency.

### 2.3. Confocal Microscopy

To assess the potential nuclear translocation of MbovP467 (WP_013954821.1) in BoMac cells, the sequence of the MbovP467 gene was custom synthesized by Beijing Tianyi Huiyuan Bioscience & Technology Inc. (Wuhan, China) and cloned into the pEGFP (Novagen, Darmstadt, Germany) vector to obtain rpEGFP-MbovP467. Following the transformation of *E. coli* strain BL21 (TransGen, Beijing, China) with the constructed plasmids (rpEGFP-MbovP467) and control plasmid (pEGFP), the rpEGFP-MbovP467 and control pEGFP plasmid were purified using the EndoFree Plasmid Kit (Omega Bio-tek, Inc. Norcross, GA, USA). The resulting rpEGFP-467 and control pEGFP plasmid were then transfected into BoMac cells in accordance with the jetPRIME^®^ (Polyplus, Saint Priest, France) protocol. In summary, each rpEGFP-MbovP467 and control pEGFP plasmid, amounting to 2 µg, were diluted in a 200 µL jetPRIME^®^ buffer. Subsequently, 4 µL of a jetPRIME^®^ reagent was added, and the resulting transfection mixture, totaling 200 µL, was added to the BoMac cells. Following a 24 h incubation period, the cells underwent fixation using 4% paraformaldehyde. Nuclear counterstaining was then carried out by applying 4′,6-diamidino-2-phenylindole (DAPI) from Beyotime (Shanghai, China) at a concentration of 5 mg/mL. A subsequent observation was conducted using a confocal microscope.

### 2.4. Gene Cloning, Expression of the Recombinant Proteins, and Polyclonal Antibody Production

The MbovP467 gene was amplified using the rpEGFP-MbovP467 plasmid as a template and specific forward and reverse primers with BamH I and Hind III restriction sites. The resultant amplicon (MbovP467) was then cloned into the pET-30a expression vector to obtain recombinant pET-30a and then transformed into *E. coli* strain BL21. The transformed *E. coli* strain BL21 was grown in an LB medium until the optical density (OD600) reached 0.6. Then, isopropyl β-D-1-thiogalactopyranoside (IPTG), at a concentration of 0.4 mM, was added to facilitate the expression of the MbovP467 protein. The purification of the MbovP467 protein was accomplished through nickel affinity chromatography (GE Healthcare, Chicago, IL, USA), following previously established procedures. A mouse antiserum against the rMbovP467 protein was generated using a standardized method. In brief, 4-week-old female BALB/c mice were purchased from the Hubei Provincial Center for Disease Control and Prevention (Wuhan, China) and housed in the Animal Facility of the Huazhong Agriculture University. The immunization protocol involved administering subcutaneous injections of 100 µg of purified rMbovP467 protein. Prior to injection, the protein was emulsified with Freund’s complete adjuvant (Sigma Aldrich, Burlington, MA, USA) for priming immunization and Freund’s incomplete adjuvant for subsequent boosters, administered at 2-week intervals. The serum was collected when antisera reached peak titers and was then stored at −20 °C for future use.

### 2.5. MbovP467 Location in BoMac Cells Detected with Western Blotting Assay

To further investigate the nuclear translocation of MbovP467 and to validate the findings obtained from the confocal microscopy, the BoMac cells were exposed to either wild-type *M. bovis* HB0801 or the mutant T5.141 strain at a multiplicity of infection of 500 for 12 h, 24 h, and 36 h. Then, the proteins from the nuclear and cytoplasm were extracted from infected cells using the MinuteTM Kit (Invent, Beijing, China), according to the provided protocol. The subcellular localization of MbovP467 was assessed through Western blotting using antisera against the rMbovP467 protein, α-tubulin (Abcam, Boston, MA, USA), or PARP (Abcam). α-tubulin and PARP served as markers for cytosolic and nuclear fractions, respectively, with anti-mouse and anti-rabbit antibodies utilized as secondary antibodies (Abcam).

### 2.6. Transcriptomic Analysis

#### 2.6.1. RNA Extraction, cDNA Library Preparation, and Sequencing

To examine how MbovP467 affects BoMac cell transcriptomes, we transfected the BoMac cells with either the rpEGFP-MbovP467 plasmid (treatment groups) or its control pEGFP plasmid (control groups). The RNA was then extracted from the treated (*n* = 3) and control cells (*n* = 3) using a TRIzol reagent (Invitrogen, Carlsbad, CA, USA). After assessing the RNA concentration and quality, the mRNA was isolated, fragmented, and used for cDNA synthesis, following the TruSeq RNA Library Preparation Kit v2 protocol (Illumina, San Diego, CA, USA). PCR amplification was performed on appropriately sized fragments, and the resulting libraries were quantified and assessed for quality. Sequencing was carried out on an Illumina HiSeq™ 2000 platform.

#### 2.6.2. Data Processing

Following sequencing, raw reads were generated and subjected to quality control processing. Reads with quality values equal to or below 10 and those contaminated with adapters or featuring a high content of unknown bases (N) exceeding 5% were systematically filtered out to obtain a set of clean reads. These high-quality reads were subsequently assembled into contigs utilizing the trinity program [20], employing parameters such as a minimum contig length set to 150 and a minimum k-mer coverage set to 3. To further refine the dataset, gene family clustering was executed using TGICL [21], ultimately yielding the final set of unigenes. The bovine genome was used as a reference genome (Table 1 and Table 2).

#### 2.6.3. DEGs Detection and Functional Enrichment Analysis

Following assembly, the assessment and quantification of gene expression levels were conducted utilizing the fragments per kilobase of exon model per million mapped reads (FPKMs) metric with RSEM v1.2.12 [22]. Differentially expressed genes (DEGs) between the control group (pEGFP) and the treatment group (rpEGFP-MbovP467) were identified using the DESeq2 version 1.30.1 [23] software and unigene expression data. The analysis relied on a negative binomial distribution to statistically evaluate raw counts. DEGs were defined based on a fold change of ≥1.0 and a *p* value adjusted to <0.05. To gain insights into the functional significance of the identified DEGs, gene ontology (GO) and Kyoto Encyclopedia of Genes and Genomes (KEGGs) analyses were employed [24]. The enrichment was determined using Fisher’s exact test, considering *p* < 0.05 as indicative of significance. The methodology for the KEGG pathway enrichment analysis followed similar principles, contributing to a comprehensive understanding of the biological processes affected by the identified DEGs.

#### 2.6.4. Real-Time PCR for mRNA Expression Pro-Inflammatory Cytokines

To investigate the inflammatory response induced by MbovP467, the BoMac cells were transfected with rpEGFP-MbovP467 and the control pEGFP plasmid. Total RNA extraction was carried out using the Trizol reagent. The quality and quantity of the extracted RNA were evaluated prior to cDNA synthesis, which was performed using the FastKing RT Kit (Tiangen Biotech Co., Ltd., Beijing, China). For qPCR, a 20 µL reaction mixture was prepared. This mixture included 0.5 µL of each primer (1.5 pmol/µL), targeting IL-1β, IL-6, IL-8, TNF-α, and Caspase 3, 2, 8, and 9; along with 7 µL of nuclease-free water; 2 µL of cDNA; and 10 µL of a SYBR Green Master mix (Vazyme Biotech Co., Ltd., Nanjing, China). The relative expression levels of genes were calculated using the ∆∆Ct method proposed by Livak and Schmittgen [25].

The primer sequences utilized in this study are detailed in Table 3.

### 2.7. Effects of rMbovP467 on the Promoter Region of TNF-α

To assess the modulatory effects of rMbovP467 on the promoter region of the TNF-α gene, a 1500 bp portion of the TNF-α gene promoter region was cloned into the pGL3-basic vector (provided by associate professor, Zhang Guangzhi, Institute of Animal Sciences, CAAS). Using cattle genomic DNA as a template and specific primers with KpnI and XhoI restriction enzyme sites, the 1500 bp promoter region was amplified. Subsequently, the amplicon was cloned into the pGL3-basic vector, resulting in the creation of the recombinant pGL3-promoter-Luc- TNF-α vector, which was then verified by PCR and sequencing. In the experimental procedures, the BoMac cells were transfected with the pGL3-promoter-Luc-TNF-α vector and were then treated with rpEGFP-MbovP467 and its blank vector pEGFP as a control group. Subsequently, the bioluminescence was measured according to the kit’s instructions (Pierce™ Firefly Luciferase Glow Assay Kit, Thermo Fisher), employing a luminescence microplate reader. This experiment was repeated three times in triplicate.

### 2.8. Assays on BoMac Cell Viability

To see the effect of rMbovP467 on BoMac cell viability, the cells were seeded in 96-well plates at a density of 500 cells per well and incubated overnight at 37 °C. Then, the cells were infected with the wild-type *M. bovis* strain HB0801, mutant T5.141, or were transfected with rpEGFP-MbovP467 for varying time intervals, with cells treated with a phosphate buffer saline (PBS) serving as the negative control. After this, 10 μL of a cell counting kit-8 reagent was added to each well, and the plates were kept in an incubator set at 37 °C for 2 h. The absorbance at 450 nm was then measured. This experiment was repeated three times in triplicate. The relative cell viability was calculated using the following formula:Relative cell viability (%) = (OD _sample_ − OD _blank)_/(OD_NC_ − OD _blank_) × 100

### 2.9. Assay on BoMac Cell Apoptosis

To investigate whether MbovP467 can induce apoptosis in BoMac cells, qPCR and flow cytometry techniques were employed. The BoMac cells were infected with the wild-type *M. bovis* strain HB0801 or mutant T5.141 and transfected with rpEGFP-MbovP467, while cells treated with PBS served as controls. Following this, the total RNA was extracted for qPCR to quantify the mRNA levels of Caspase 2, 3, 8, and 9. For the flow cytometry analysis, the treated cells were stained using the Annexin V-FITC Apoptosis Detection Kit (Vazyme), according to the manufacturer’s recommendations. Subsequent to staining, flow cytometry was performed for the detection of apoptosis induced by MbovP467, and the acquired data were analyzed using the FlowJo software version 10.8.1 (FACSVerse, Becton Dickinson, NJ, USA). This experiment was repeated three times in triplicate.

### 2.10. Statistical Analysis

The presented data were expressed as means ± standard error of the mean (SEM). The statistical analysis was performed using Student’s *t*-test for single comparisons and a one-way analysis of variance (ANOVA) for multiple comparisons, utilizing the GraphPad Prism version 5 software (GraphPad Software, La Jolla, CA, USA).

## 3. Results

### 3.1. Nuclear Localization of MbovP467

The protein sequence of the monocistronic gene MbovP467 was retrieved from National Center for Biotechnology Information database (NCBI) (WP_013954821.1). According to ProtParam, MbovP467 has a molecular weight of 77.2 kDa and an isoelectric point of 8.94. A prediction signal peptide (Sec/SPI) was predicted between residues 27 and 28 aa using SignalP 5.0 (Appendix A). A prediction nuclear localization signal (NLS) sequence was predicted using cNLS Mapper (cut-off score of ≥5) (Appendix A). After searching the NCBI database for homologs of MbovP467, three DUF285 domains were found between residues 322–433 aa, 440–553 aa, and 513–655, respectively. Furthermore, a comparison of the DUF285 motifs of MbovP467 and MbovP475 (another nucleomodulin of the *M. bovis* strain HB0801) showed several conserved amino acid sites (Figure 1).

Based on the predicted presence of NLS, it was hypothesized that MbovP467 possesses signals facilitating nuclear entry. To experimentally confirm this, the rpEGFP-MbovP467 was transfected into the BoMac cells alongside the pEGFP vector for control purposes, and the cells were subsequently observed under a confocal microscope. The findings show punctuated green fluorescence in both the nuclei and cytoplasm of the cells transfected with rpEGFP-MbovP467. In contrast, the BoMac cells transfected with the control plasmid displayed diffuse green fluorescence within the cellular cytoplasm and nuclei (Figure 2). This microscopy-based evidence supports the assertion that MbovP467 harbors NLS, facilitating its entry into the cellular nuclei.

### 3.2. Confirmation of Nuclear Localization of MbovP467 with Western Blotting

The cytoplasmic and nuclear proteins were extracted at 12 h, 24 h, and 36 h from the BoMac cells infected with the wild-type *M. bovis* strain HB0801 or mutant T5.141 for a Western blotting assay. PARP and α-tubulin were used as protein targets for primary antibodies for nuclear and cytoplasmic markers, respectively, and mouse antisera against rMbovP467 were used for MbovP467. As expected, the band for PARP was only located in the nuclei, while α-tubulin was confined to the cytoplasm. The MbovP467 protein from the wild-type *M. bovis* strain HB0801-infected BoMac cells was detected in both the nuclei and cytoplasm. In contrast, no band corresponding to MbovP467 was observed in the case of the mutant T5.141 (Figure 3). These pieces of evidence support the confocal microscopy results and suggest the nuclear entry of secreted MbovP467.

### 3.3. Analysis of Differentially Expressed Genes of Transcriptome

RPKM was employed as a metric to estimate gene expression levels in the BoMac cells transfected with the rpEGFP-MbovP467 plasmid (treatment groups) and its blank vector pEGFP (control groups), respectively. The RPKM density map (Figure 4A) provides an overview of the gene expression patterns across the entire dataset. The volcano map (Figure 4B) visually represents the differential expression of the 240 genes (Appendix A), with 220 genes exhibiting up-regulation (Appendix A) and 20 genes displaying down-regulation (Appendix A). A hierarchical clustering analysis of the DEGs highlighted high reproducibility within the sample groups, with distinctions between the groups depicted in red and blue (Figure 4C).

### 3.4. GO and KEGG Enrichment of Differential Expressed Genes

The GO analysis provided comprehensive annotations for DEGs in terms of molecular function, biological process, and cellular component. This approach facilitated a holistic understanding of potential functions associated with the observed gene expression profiles. The enriched GO annotation terms for differentially expressed genes are visually presented in Figure 5. This analysis revealed the identification of 240 differentially expressed genes of which 220 were up-regulated and 20 down-regulated. These differentially expressed genes encompassed those involved in inflammatory cytokine receptor binding, neutrophil activation, the regulation of cell proliferation, inflammatory responses, the chemokine-mediated signaling pathway, apoptotic processes, the G protein-coupled receptor signaling pathway, and nuclear receptor activities. Additionally, the KEGG database was leveraged for the enrichment analysis of biological pathways associated with the identified gene collections. DEGs were systematically categorized into various biological pathways, allowing for a comprehensive analysis of the impact and regularity of functional variations on these pathways. The KEGG signal pathways were organized into five overarching categories based on BRITE (Biomolecular Relations in Information Transmission and Expression) [26] hierarchies, including cellular processes, environmental information processing, genetic information processing, and metabolism. The KEGG pathway enrichment unveiled that the DEGs primarily participated in micro RNAs in cancer, pathways in cancer FoxO signaling, the C-type lectin receptor signaling pathway, the cytokine–cytokine receptor signaling pathway, the HF-1 signaling pathway, the IL-17 signaling pathway, the NF-κB signaling pathway, and the TNF signaling pathway (Figure 5).

### 3.5. RNA-Seq Data Validation

#### 3.5.1. Effects of rMbovp467 on mRNA and Promoter Region of Pro-Inflammatory Cytokines

The mRNA levels of IL-1β, IL-8, and TNF-α were significantly higher in the BoMac cells transfected with rpEGF-MbovP467 as compared to the BoMac cells transfected with the blank vector pEGFP, indicating a potential inflammatory role of MbovP467 in *M. bovis* pathogenesis. Further, we tested the effect of MbovP467 on the promoter region of TNF-α, in which the reporter plasmid pGL3-promoter-Luc-TNF-α containing the 1500 bp promoter region of TNF-α was transfected in the BoMac cells which were then transfected with rpEGFP-MbovP467 and the control pEGFP plasmid. A significant difference in the promoter activity of TNF-α was noticed between rpEGFP-MbovP467 and the control pEGFP group, showing a stimulatory effect of the nucleomodulin MbovP467 on the activity of the TNF-α promoter region (Figure 6).

#### 3.5.2. MbovP467 Reduces BoMac Cell Viability

Our investigation into the effect of MbovP467 on cell viability revealed a significant difference in relative cell viability, with cells infected with T5.141 showing a notably higher viability compared to those infected with the wild-type *M. bovis* strain HB0801 or rpEGFP-MbovP467 at 24 h, 36 h, and 46 h post-infection. However, no significant difference in relative cell viability was observed between the wild-type *M. bovis* strain HB0801 and rpEGFP-MbovP467 (Figure 7).

#### 3.5.3. MbovP467 Induces Apoptosis in BoMac Cells

Our qPCR and flow cytometry analyses revealed that MbovP467 induced apoptosis in BoMac cells. A statistically significant difference (*p* < 0.05) was observed in the mRNA levels of Caspase 2, 3, 8, and 9 between the wild-type *M. bovis* strain HB0801 and the mutant T5.141. Similarly, a significant difference (*p* < 0.05) was noted in the mRNA levels of Caspase 2, 3, 8, and 9 between the mutant T5.141 and rpEGFP-MbovP467. Additionally, a significant difference was observed in the mRNA levels of Caspase 2, 3, 8, and 9 between the wild-type *M. bovis* HB0801 and rpEGFP-MbovP467 (Figure 8). Flow cytometry was used to further confirm apoptosis induction in the BoMac cells infected by the wild-type *M. bovis* strain HB0801, the mutant T5.141, or transfected with rpEGFP-MbovP467 after staining with annexin V and propidium iodide. The apoptosis levels induced by the wild-type *M. bovis* strain HB0801, the mutant T5.141, or rpEGFP-MbovP467 and PBS (NC) were 30.60 ± 2.00%, 9.00 ± 2.00%, 16.50 ± 1.52%, and 2.21 ± 0.70%, respectively. A significant difference in apoptosis levels between the wild-type *M. bovis* HB0801 strain, the mutant T5.141, rpEGFP-MbovP-467, and the control group was observed (Figure 9).

## 4. Discussion

Over the past decade, researches have increasingly uncovered the ability of bacterial pathogens in mammals and plants to directly target cellular nuclei by their nucleomodulin proteins. Nucleomodulins are secreted proteins of bacteria that enter into the nuclei of host cells and control the expression of crucial genes involved in important host responses, such as immunity, cell proliferation, and apoptosis [11,20,27,28,29,30,31,32,33]. However, the exploration of the secreted nucleomodulins of mycoplasmas is still in its early stages. Previous evidence regarding mycoplasmas suggested that the secreted DNA methyltransferases of *M. hyorhinis* and the nucleomodulin MbovP475 of *M. bovis* can enter into nuclei of host cells and subvert cell transcriptional profiles leading to proliferation changes and apoptosis [14,15]. The present investigation establishes MbovP467 as a novel nucleomodulin in *M. bovis*, which shows homologies with other proteins of other strains of *M. bovis*, including the MBOVPG45 0425, MBOVPG45 0418, MBOVPG45 0419, and MBOVPG45 0509 proteins of the *M. bovis* PG45 strain. Further, MbovP467 also shows homologies with proteins present in *M. feriruminatoris*, *M. leachii*, and *M. mycoides* subsp. *capri* (Appendix A). To see whether MbovP467 can enter into the nucleus of BoMac cells, the sequence of the MbovP467 gene was cloned into the pEGFP vector to obtain rpEGFP-MbovP467, which was then transfected into the BoMac cells, along with its control plasmid. The confocal microscopy results show a punctuated green color in rpEGFP-MbovP467, whereas the green color was widely distributed in the case of the control EGFP plasmid. This confocal microscopy observation supports the fact that MbovP467 has an NLS, and as EGFP does not have NLS, the entry of EGFP into the nucleus without a nuclear localization signal (NLS) can occur through diffusion [34]. Although EGFP can enter the nucleus without a typical NLS, the observed nuclear localization of the MbovP467-EGFP fusion protein via confocal microscopy does not solely confirm the nuclear localization of the MbovP467 portion. So, in the next step, we performed Western blotting to confirm the nuclear entry of Mbovp467 using antisera directed against MbovP467. Our Western blotting results confirm the nuclear entry of MbovP467.

MbovP467 exerts regulatory influence over key genes implicated in pivotal host responses, including those related to inflammation, cell viability, and apoptosis, as evidenced by our comprehensive analysis by a cell viability assay, apoptosis assay, and RNA-seq data. Our cell viability assay findings reveal significant differences in relative cell viability between the wild-type *M. bovis* strain HB0801, mutant T5.141, and rpEGFP-MbovP467-infected cells, which are consistent with our research objectives. However, we did not use the fusion of irrelevant proteins with EGFP, because the use of rpEGFP-MbovP467 serves as an effective tool for investigating the specific role of MbovP467 in cell viability, as evidenced by the observed differences compared to the wild-type strain. While the inclusion of a control protein could provide additional context, our experimental design aligns with established methodologies [14] in the field, and the results contribute meaningfully to the current body of knowledge.

The RNA-seq analysis unveiled 240 DEGs, with 220 up-regulated and 20 down-regulated ones under the influence of MbovP467. Notably, these DEGs displayed enrichment in critical signaling pathways, such as the TNF signaling pathway, the NF-kappa B signaling pathway, the C-type lectin receptor signaling pathway, the cytokine–cytokine receptor signaling pathway, the G protein-coupled receptor signaling pathway, the chemokine-mediated signaling pathway, the FoxO signaling pathway, nuclear receptor activities, sequence-specific DNA binding, and pathways in cancer. These pathways are closely associated with different important biological processes. For example, the TNF/NF-κB signaling pathway is a well-established inflammatory cascade with implications in various aspects of bacterial pathogenesis. The activation of this pathway is a fundamental component of the host immune response to bacterial infections [35]. Macrophages are the primary producers of molecules belonging to the TNF superfamily, with TNF-α standing out as a potent pro-inflammatory cytokine crucial for immune function, inflammation, and the regulation of cell growth, differentiation, and apoptosis [36,37]. NF-κB, a homodimeric and heterodimeric complex comprising Rel family members (NF-κB1, NF-κB2, RelA, RelB, and c-Rel), which also exist in *Bos taurus*, regulates the expression of various genes involved in immune response and diverse cellular processes, including growth, development, and apoptosis [38,39]. The G protein-coupled receptors regulate the expression of several downstream signaling pathways with a variety of biological actions, including cell migration, proliferation, and apoptosis [40,41,42]. Similarly, nuclear receptor activity proteins act as transcription factors located in the cell nucleus, bind to specific DNA sequences, and modulate the transcription of nearby genes, thereby regulating various physiological processes, such as development, metabolism, immune response, and cell proliferation [43]. Sequence-specific DNA-binding proteins selectively bind to specific DNA sequences typically found in gene promoter regions and play a crucial role in controlling gene expression, either activating or repressing transcription [44]. Our findings align with prior investigations on nucleomodulins from diverse bacterial species, highlighting their role in manipulating host cellular responses. For instance, the nucleomodulin Ank13 from *Orientia tsutsugamushi* exploits the RaDAR nuclear import pathway to regulate host cell transcription [32]. Similarly, the nucleomodulins VirE3 and HsvG from *Agrobacterium* and *Pantoea agglomerans* translocate to host cell nuclei, modulating the expression of various genes [45,46]. The nucleomodulin LntA from *Listeria monocytogenes* stimulates innate immune responses by inhibiting the chromatin-associated repressor BAHD1 [47]. Additionally, the nucleomodulins HP0425 and HP0059 from *Helicobacter pylori* translocate into host cell nuclei, contributing to gastric cancer [29,48]. Likewise, the nucleomodulin SspH1 from *Salmonella* enters the host cell nucleus, inhibiting NF-κB-dependent genes [49]. Furthermore, comparable findings have been documented regarding nucleomodulins in *M. hyorhinis*. These nucleomodulins, acting as methyltransferases, translocate into host cell nuclei, thereby altering the epigenetic landscape through DNA methylation, which results in the modulation of genes in cancer [12].

Our qPCR results are in agreement with the transcriptomic data, revealing that MbovP467 enhances the expression of inflammatory cytokines, contributing to the inflammation observed during *M. bovis* infection. Consistent inflammatory responses during *M. bovis* infection have been documented by various researchers [50,51,52,53]. To further elucidate the role of the novel nucleomodulin MbovP467 in *M. bovis*-induced inflammation, we examined its effect on the TNF-α promoter region. Our findings indicate that MbovP467 increases the activity of the TNF-α promoter region. Additionally, our cell viability and apoptosis assays, conducted through flow cytometry and qPCR, substantiate the RNA-seq data, confirming that MbovP467 diminishes BoMac cell viability and induces apoptosis. This could possibly be due to the modulation of antiapoptotic, apoptotic, and or inflammatory genes by MbovP467, akin to the findings of Zhao et al. (2022), who demonstrated that the nucleomodulin MbovP475 from the same *M. bovis* strain (HB0801) attenuates cellular viability through the modulation of CRYAB and MCF2L2 expressions [14]. Further, this could also be due to the modulation of TNF-α as TNF-α is a vigorous pro-inflammatory cytokine with a crucial role in the immune function, inflammation, and regulation of cell growth, differentiation, and apoptosis [37]. However, further investigation is needed to unveil *M. bovis*-induced inflammation, cell viability, and apoptosis, such as ChiP-seq to identify the specific host DNA-binding regions of MbovP467 or to assess the modulation of histone proteins through acetylation or methylation, as well as host DNA methylation. Previous studies on nucleomodulins from bacteria have reported interactions with host DNA methylation, histone acetylation, and methylation [11,12,33,47,54,55,56,57].

## 5. Conclusions

This study reveals a novel nucleomodulin MbovP467 of *M. bovis*. A transcriptomic analysis highlighted significant gene expression changes in key pathways related to inflammation, apoptosis, and signalling. Experimental assays demonstrated that MbovP467 reduces cell viability and induces apoptosis in BoMac cells, accompanied by an up-regulation of pro-inflammatory cytokines and an enhancement of the promoter activity of TNF-α. These findings indicate that MbovP467 would likely contribute to *M. bovis* pathogenesis.

## Figures and Tables

**Figure 1 cells-13-00604-f001:**
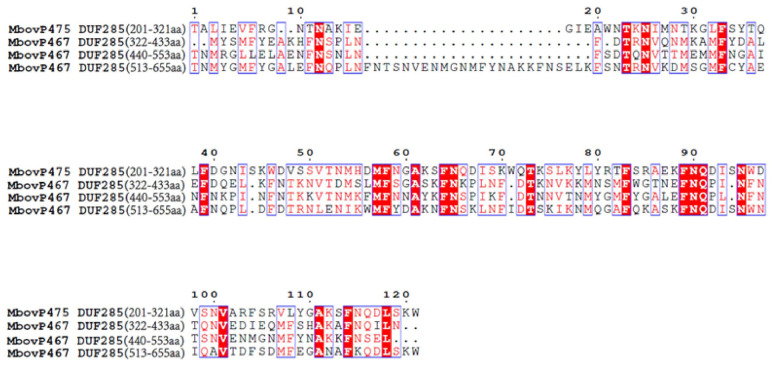
After searching the NCBI database for homologs of MbovP0145, three DUF285 domains were found between residues 322–433 aa, 440–553 aa, and 513–655, respectively. Furthermore, a comparison of the DUF285 domains of MbovP467 and MbovP475 (another nucleomodulin of the *M. bovis* strain HB0801) showed several conserved amino acid sites.

**Figure 2 cells-13-00604-f002:**
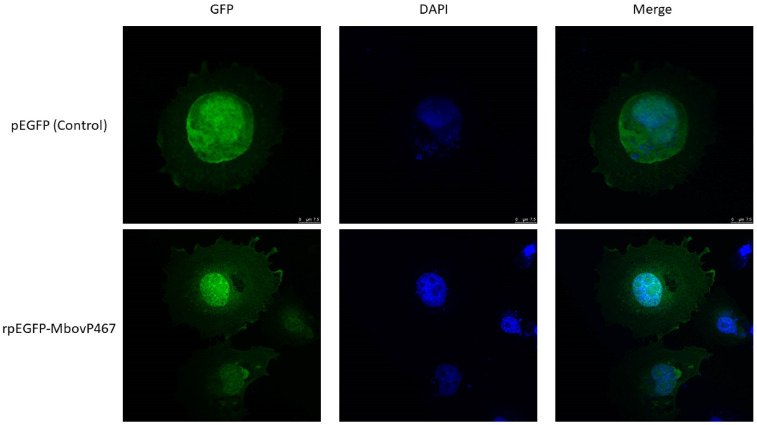
An observation of the nuclear localization of MbovP467 was conducted using a confocal microscope. The cytoplasm and nuclei of cells transfected with rpEGFP-MbovP467 showed punctuated green fluorescence. Meanwhile, diffuse green fluorescence was exhibited from the nuclei and cytoplasm of the cells transfected with control plasmids.

**Figure 3 cells-13-00604-f003:**
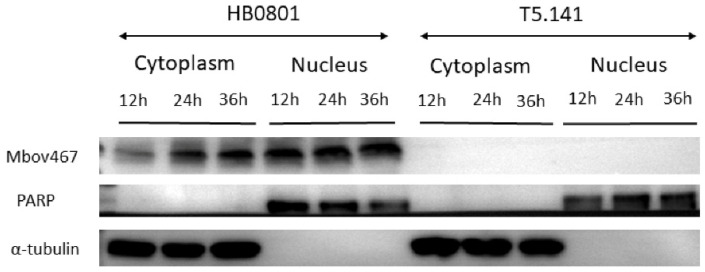
Western blotting was used to validate the confocal microscopy finding and to further investigate the nuclear entry of MbovP467. The bands were observed from both nuclear and cytoplasmic proteins extracted from the wild-type *M. bovis* HB0801-infected BoMac cells. In contrast, no bands were observed from the nuclear and cytoplasmic protein extracted from the mutant T5.141-infected BoMac cells. α-tubulin (cytoplasmic marker) only exhibited bands in the cytoplasmic protein, and PARP (nuclear marker) only exhibited bands in the nuclear protein.

**Figure 4 cells-13-00604-f004:**
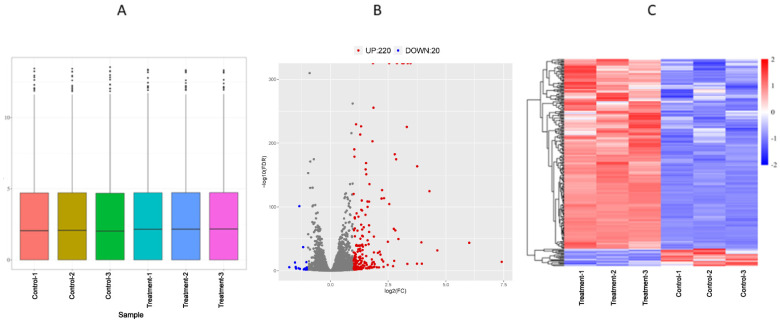
Analysis of DEGs in BoMac cells. (**A**) Utilization of the reads per kilobase per million mapped reads (RPKMs) density map to scrutinize the comprehensive gene expression patterns of the entire sample. (**B**) Generation of a volcano plot depicting the differential expression of genes within BoMac cells. (**C**) Implementation of hierarchical clustering analysis to delineate the patterns of differentially expressed genes.

**Figure 5 cells-13-00604-f005:**
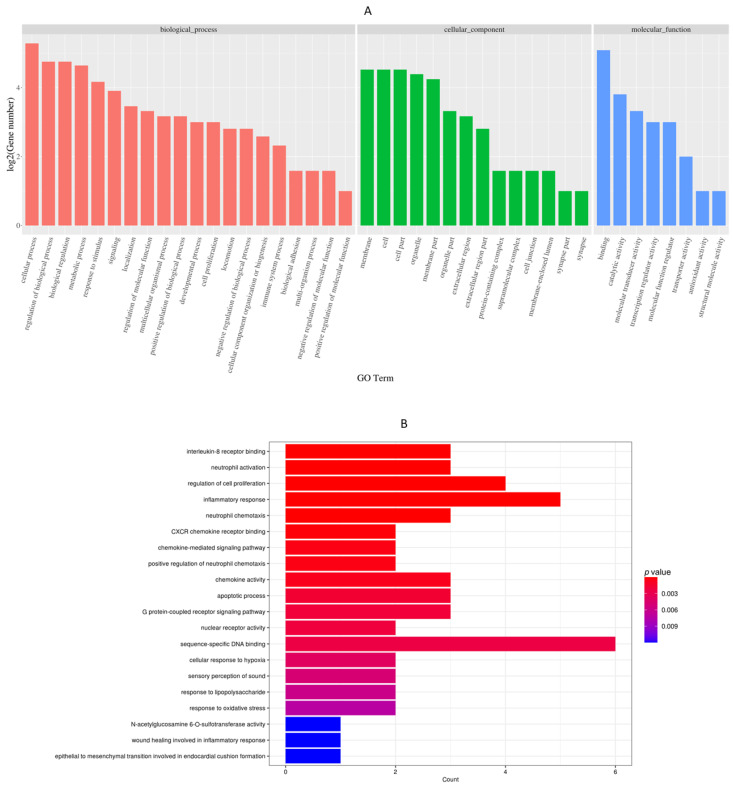
Analysis of DEGs profiles. (**A**) Examination of GO terms related to DEGs. (**B**) Functional analyses illustrating the enrichment of biological functions associated with DEGs. (**C**) Classification based on the KEGG for DEGs. (**D**) Enriched functional analysis providing insights into the functional implications of DEGs. The GO enrichment analysis reveals an enrichment of genes associated with *Legionella*, *Leishmania*, and pertussis. It is important to note that these pathogens are typically not associated with cattle. The presence of these genes in the results stems from the algorithms used in the analysis, which include a broad range of gene annotations.

**Figure 6 cells-13-00604-f006:**
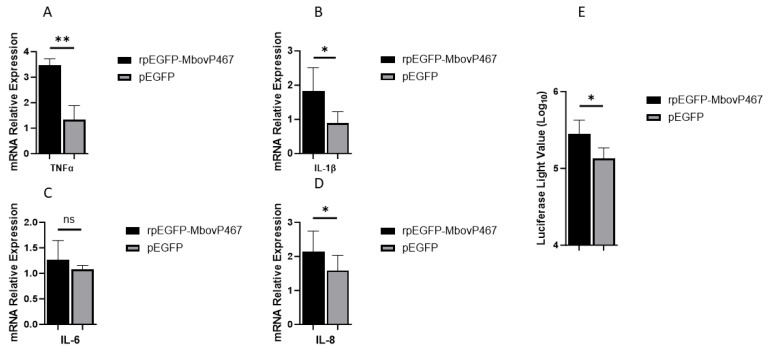
mRNA expression of inflammatory cytokines TNF-α (**A**), IL-1β (**B**), IL-6 (**C**), IL-8 (**D**) and effect of rpEGFP-MbovP467 on TNF-α promoter region (**E**). * *p* < 0.05 and ** *p* < 0.01 indicate statistically significant differences; ns, indicates no difference.

**Figure 7 cells-13-00604-f007:**
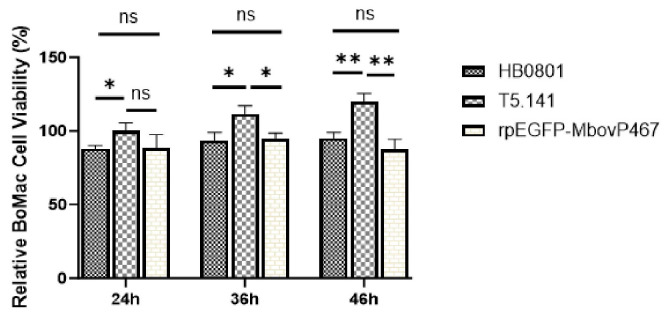
MbovP467 impact on BoMac cell viability. The assessment of BoMac cell viability was conducted using the Cell Counting Kit-8 (CCK-8). Relative cell viability in BoMac cells infected with the wild-type *M. bovis* HB0801, mutant T5.141 infection, and rpEGFP-MbovP467 revealed a significant decrease in cell viability attributed to the presence of MbovP467. * *p* < 0.05 and ** *p* < 0.01 indicate statistically significant differences; ns, indicates no difference.

**Figure 8 cells-13-00604-f008:**
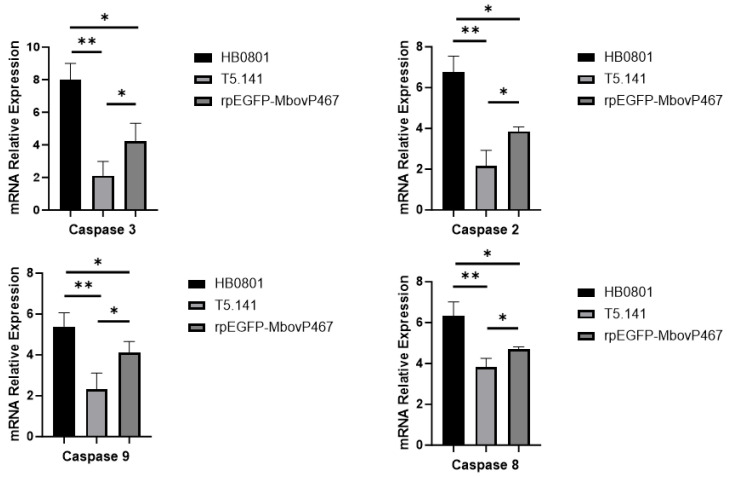
Relative mRNA expression of Caspase 3, 2, 8, and 9 in BoMac cells during wild-type *M. bovis* HB0801, mutant T5.141 infection, and rpEGFP-MbovP467 treatment. * *p* < 0.05 and ** *p* < 0.01 indicate statistically significant differences.

**Figure 9 cells-13-00604-f009:**
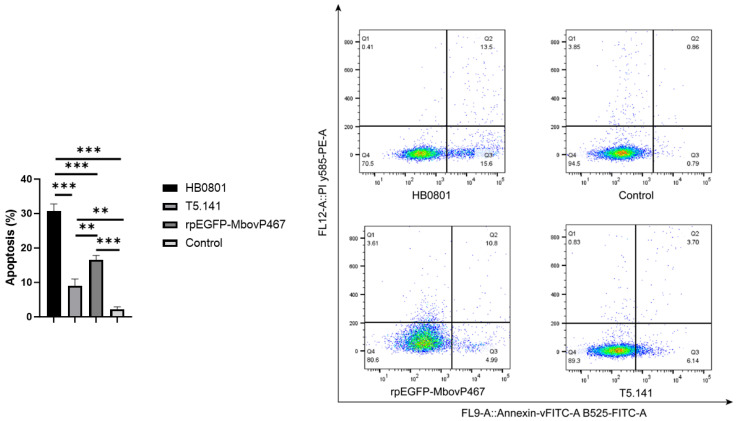
Apoptosis of BoMac cells during wild-type *M. bovis* strain HB0801 and mutant T5.141 infection or transfection with rpEGFP-MbovP467 by flow cytometry. For flow cytometry, 5 × 10^5^ BoMac cells/wells were seeded in a six-well plate and grown at 37 °C overnight. They were then infected with the wild-type *M. bovis* strain HB0801, mutant T5.141, or transfected with rpEGFP-MbovP467. After treatment, cells were stained with the Annexin V-FITC Apoptosis Detection Kit and analyzed using the FlowJo software version 10.8.1 (FACSVerse, USA). ** *p* < 0.01 and *** *p* < 0.001 indicate statistically significant differences.

**Table 1 cells-13-00604-t001:** Clean data statistics and summary of filter data.

Samples	Raw Reads	Raw Bases	Clean Reads	Clean Bases	Clean Ratio	Q20	Q30	GC
Control 1	59,992,918	8,998,937,700	59,543,994	8.92 × 10^9^	99.25%	98.04%	94.35%	48.33%
Control 2	42,413,822	6,362,073,300	42,136,802	6.31 × 10^9^	99.35%	98.26%	94.83%	48.37%
Control 3	58,935,792	8,840,368,800	58,511,866	8.76 × 10^9^	99.28%	98.08%	94.42%	48.35%
Treatment 1	83,245,064	12,486,759,600	82,730,332	1.24 × 10^10^	99.38%	98.35%	95.10%	48.48%
Treatment 2	70,364,650	10,554,697,500	69,876,584	1.05 × 10^10^	99.31%	98.15%	94.61%	48.46%
Treatment 3	69,921,538	10,488,230,700	69,479,750	1.04 × 10^10^	99.37%	98.27%	94.92%	48.65%

Raw data statistics include information about individual reads, with each set of four consecutive lines providing details about one read. The total number of reads per file was then calculated. Raw base statistics encompass the overall count of bases in the raw data. Clean reads were obtained by filtering the original data—eliminating linker sequences, discarding contaminated portions, and excluding sequences with an excess of low-quality bases. The GC% represents the proportion of G and C bases in the entire dataset. The accuracy of base calls was reflected in Q20 and Q30, denoting the percentages of bases with Phred scores exceeding 20 and 30, respectively. Clean base statistics involve determining the count and length of sequences within clean reads, thereby calculating the total number of clean bases. The clean bases% was calculated as the percentage of clean bases relative to the total raw bases. Note: Treatment 1, treatment 2, and treatment 3 represent the BoMac cells transfected with the rpEGFP-MbovP467 plasmid (treatment groups), and control 1, control 2, control 3 represent the BoMac cells transfected with its control pEGFP plasmid (control groups).

**Table 2 cells-13-00604-t002:** Summary of alignment results.

Samples	Total Reads	Total Mapped Reads	Map Rate	Unique	Paired	Single	Self and Mate	Map Diff CHR
Control 1	59,543,994	57,467,676	96.51%	55,323,506	53,793,422	1,616,810	55,850,866	391,760
Control 2	42,136,802	40,710,940	96.62%	39,379,997	38,294,926	1,134,004	39,576,936	321,496
Control 3	58,511,866	56,444,119	96.47%	54,499,140	52,903,316	1,671,653	54,772,466	467,834
Treatment 1	82,730,332	79,971,255	96.66%	77,250,388	75,151,040	2,199,405	77,771,850	616,782
Treatment 2	69,876,584	67,520,868	96.63%	65,269,363	63,463,956	1,887,772	65,633,096	533,124
Treatment 3	69,479,750	67,058,501	96.52%	64,775,746	62,935,550	1,923,865	65,134,636	566,662

The effective reads indicate the number of clean reads that remain after eliminating rRNA reads. These reads were then used for aligning to the genome. Total mapped refers to the total count of sequencing sequences that align to the genome. Uniquely mapped indicates the count of sequencing sequences with exclusive alignment positions on the reference sequence. Note: Treatment 1, treatment 2, and treatment 3 represent the BoMac cells transfected with the rpEGFP-MbovP467 plasmid (treatment groups), and control 1, control 2, control 3 represent the BoMac cells transfected with its control pEGFP plasmid (control groups).

**Table 3 cells-13-00604-t003:** List of primers used in this study.

Genes	Forward Sequence	Reverse Sequence
Caspase 3	ACCTCTTCTGCCCTGACTTC	TGTAACTACTGAAGCCCGCA
Caspase 2	TTTCTCCCTGCTCACCTCAG	TCTGCCTTCATACTGTGCCA
Caspase 8	TTCCTTTGCTGCCTCGAGTA	GTTGGGAGTGGGGAGATTCA
Caspase 9	CTCCTCTCCTTTTGCCCTCA	TGGACCATAAAGCAGGCTGA
IL-1β	TCCGACGATTTCTGTGTTGA	ACGAGGAGGCCAAGAAAAGA
IL-6	GGTCAAGATGCCAAGTCAGC	TGGGATCTTCACTGAATGCT
IL-8	ACTGTGTGGGTCTGGTGTAG	AGGCGAGGGTTGCAAGATTA
TNF-α	TTGTTCCTCACCCACACCAT	CCAAAGTAGACCTGCCCAGA

## Data Availability

The data is provided in the article.

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
