# Peer review of "The Identification of a Novel Nucleomodulin MbovP467 of Mycoplasmopsis bovis and Its Potential Contribution in Pathogenesis"

_cells, 2024, doi:10.3390/cells13070604_

Round 1

Reviewer 1 Report

Comments and Suggestions for Authors

The paper with the title “Identification of a novel nucleomodulin MbovP467 for Mycoplasma bovis and its potential contribution in pathogenesis” is an impressive one. The study identified a new nucleomodulin MbovP467 for M. bovis which might have an important role in M. bovis pathogenesis. The quality of it is great. It is well written and very few things can be said to improve it. A lot of laboratorial and bioinformatics work has been done and the results are unquestionable.

Specific comments:

L73: typified by or typified as?

L94: Did you add animal serum? In what percentage?

L95: Add here the information that appears in L 143-144 (Information about the mutant MbovP467 knockout T5.141 strain screened from the mutant library 143 previously constructed by this laboratory [12].)  And in L143-144, just mention the mutant without further information.

Table 1 and 2, Figure 3…: I would prefer to use the word Control instead of Con. This can be applied to the rest of the paper.

Figure 2: I suggest to merge figures 1A and 2B.

L330: M. bovis in cursive

L331: treated instead of treatment

Figure 3: The font size of the images should be increase. It is difficult to read.

Figure 4: Add the corresponding capital letter to each image: A, B, C and D

Figure 8: The flow cytometry image needs to be explain in the title of the figure. And add capital letters for the two images in the figure.

L446: mycoplasmas instead of Mycoplasmas.

L474: nuclear instead of Nuclear.

Author Response

Comment 1

L73: typified by or typified as?

Response 1

Thank you for pointing this out. It has been corrected, we re-wrote our manuscript (L 87 in revised manuscript)

Comment 2

L94: Did you add animal serum? In what percentage?

Response 2

We had added 50 % horse serum. We added this information to the revised manuscript (L 113)

Comment 3

 L95: Add here the information that appears in L 143-144 (Information about the mutant MbovP467 knockout T5.141 strain screened from the mutant library 143 previously constructed by this laboratory [12].)  And in L143-144, just mention the mutant without further information.

Response 3

Thank you for pointing this out. We agree with this comment. Therefore, we have added the information about mutant MbovP467 knockout T5.141 strain in the section “2.2. Growth of Cells and Bacterial Strains” (L114-117 in the revised manuscript), and in line 174 and throughout the revised manuscript we just mentioned the mutant T5.141 according to your suggestion.

Comment 4

Table 1 and 2, Figure 3…: I would prefer to use the word Control instead of Con. This can be applied to the rest of the paper.

Response 4

The “control” has been added in Tables 1, 2, and Figure 3, and in the rest of the revised manuscript.

Comment 5

Figure 2: I suggest to merge figures 1A and 2B.

Response 5

Thank you for pointing this out. We agree with this comment. Therefore, the western blotting results of mutant T5.141 which appeared two times in Figure 2 (A-B) have been deleted.

Comment 6

 L330: M. bovis in cursive

Response 6

  1. bovishas been italicized (L382 in the revised manuscript).

Comment 7

L331: treated instead of treatment

Response 7

Thank you for pointing this out. We deleted this result as it appeared two times and further we need to find out the mechanism of this (L383 in the revised manuscript).

Comment 8

 Figure 3: The font size of the images should be increase. It is difficult to read.

Response 8

The font size of Figure 3 has been increased.

Comment 9

 Figure 4: Add the corresponding capital letter to each image: A, B, C and D

Response 9

We appreciate your comments. We already added the corresponding capital letters (A, B, C, and D) for each image when we submitted our separate files of figures in the journal but it is not appearing in word file in the journal system. However, we added the corresponding capital letters (A, B, C, and D) for each image again.

Comment 10

 Figure 8: The flow cytometry image needs to be explain in the title of the figure. And add capital letters for the two images in the figure.

Response 10

The have explained our flow cytometry image in more detail in the revised manuscript.

Comment 11

 L446: mycoplasmas instead of Mycoplasmas.

Response 11

Thank you for pointing this out. The “Mycoplasmas” has been replaced with “mycoplasmas” (L517 in the revised manuscript).

Comment 12

 L474: nuclear instead of Nuclear.

Response 12

Thank you for your comments. The word “Nuclear” has been replaced with “nuclear” (L545 in the revised manuscript).

Reviewer 2 Report

Comments and Suggestions for Authors

In this manuscript, Raheem and colleagues assess and demonstrate potentially important activities of MbovP467 from Mycoplasma bovis. Most of the experiments are well controlled (see below for some caveats) and the results add to the field of mycoplasma species nucleomodulins.

Scientific Points

1.       It would be useful to include the currently proposed name for the taxon in Keywords, the title, the abstract and the main text. Mycoplasmopsis bovis

2.       References L47 and L50 and L53-57

3.       It would be useful to mention their obligate lifestyle as host associated

4.       L74 As written, this suggests that other species have a SAP11. Is this the intention?

5.       Other publications have identified NLS in M. hominis and I believe a nuclease for M. genitalium or M. fermentans. Other examples in the Mycoplasma should be introduced or discussed.

6.       It would be useful to give a unique protein identifier so that P467 can be retrieved from GenBank. Is it conserved in all Mb genomes? Other species? Domains or functional regions other than NLS and LP signal?

7.       L95 is this mutant by transposon or homologous recombination? Is the gene in an operon (in which case there might be polar effects that should be mentioned). Complementation would also be needed.

8.       L108 Was the lipoprotein gene constructed with or without LP signal?

9.       A figure showing the approx. location 0of NLS and the match to consensus would be helpful

10.   Please include what is a significant p value. Please only use one style of P (upper or lower, italics or not). Two different forms are found in text.

11.   L294 Was only one or more NLS found-it is written as “signals”?

12.   Section 3.1. How does EGFP enter without NLS? This should be clarified in text. Also, this is a caveat that you cannot tell whether there is additional localization of recombinant dual protein following EGFP dependent nuclear entry.

13.   How does rMbov467 protein get taken up?

14.   Fig 4 shows genes involved in Legionella (I do not think bovines get this), and pertussis (another human disease) and Leishmania (do cattle get this)? The authors have produced a Figure of what the algorithms produce, but context and curation are needed in presentation of results.

15.   L388 and L410. This is possible, bit overstated here. They have the potential to be involved in infection and pathogenesis in the animal

16.   L408 No control protein is provided—also it is better to have EGFP fused to an irrelevant protein.

17.   Fig 2 should be Fig 6 p14

18.   The legend to Fig 8 should indicate what is in the control. Is it EGFP?

19.   The Discussion starts with a very long paragraph of 1 page. I would split into multiple and include appropriate transitions.

20.   L448 are either P475, P467 MTases? This should be mentioned one way or another.

21.   It would be informative to mention unambiguously whether the many proteins L469/L470 are all known to exist in Bos taurus.

Language points

1.       Title perhaps use “of Mycoplasma” rather than “for”

2.       L21 and L29 Depending on journal style, it may not be necessary to include the abbreviated taxa names in parentheses.

3.       L24 This study does not aim to identify, but characterizes one that is identified.

4.       L22 related (non italics)

5.       L26 with a nuclear

6.       L34 responses (also L59)

7.       L35 pathways

8.       L36 processes

9.       L67 and L68 have different capitalization for Mycoplasma

10.   L95 and elsewhere  For companies, please include city and State for USA based companies and city and country for others

11.   L97 gentamicin

12.   L97 Escherichia should be in full on first use in main text

13.   L108 source or reference for pEGFP

14.   L113 please use comma rather than hyphen

15.   2.4 and 2.5 (and elsewhere) Please use one style of capitalization for headers

16.   L125 location and company are not needed as given L109

17.   L142 and throughout, wild-type rather than wild

18.   L152 please add city after Abcam…on second instance only company name is needed

19.   L153 sources of secondaries

20.   L162 company location for Agilent

21.   L169 company location for ABI

22.   L174 processing

23.   L179, L219, L221, L225 sources or references of databases and tools should be included

24.   L183 summary (lower case)

25.   Tables Since Treatment is in full, please use Control in full

26.   L187 These do not match the contractions of Cont in Table, same with Table 2 and L214

27.   L202 were aligned

28.   L203, L206, L208, L209, L210. Theare are terms and symbols (“+” for example) that do not appear in Table 2. Please check the complete footnotes

29.   L231 responses

30.   L232 or control pEGFP plasmid.

31.   L234 company location also L238

32.   L236 add caspases since these are in Table 3

33.   L238 the year is not required here

34.   L245 on the promoter

35.   L247 1500-bp portion of the TNF

36.   L247 source/ref for plasmid, Flowjo L285

37.   L248-9 is repetitive partially with earlier text, also L254

38.   L256 a control group or “control groups” Please match singular and plural

39.   L257 which kit?

40.   L263 Please define CCK-8 and PBS L267

41.   L298, L205 punctated or punctuated?

42.   L309 of Nuclear

43.   L312 data not shown

44.   L314 a western

45.   L322 protein treated

46.   Fig 2 align the hours so on same horizontal line. Please also arrange hours for panel B so that they match panel A. It is confusing to have them in opposite arrangements

47.   Fig 2 cytoplasm does not require italics

48.   L330 and elsewhere. M bovis should always be italicized (multiple examples in legends)

49.   L331 the mutant is duplicated

50.   L335 Analysis of

51.   L337 Treatment (to match elsewhere)

52.   Fig 3 Control (to match full length “treatment”)

53.   L357 are visually

54.   L359 encompassed those involved in inflammatory

55.   L367 Whate are Brite hierarchies? Please include in text

56.   P value (Figure 4 panels—please also see note about styles)

57.   L384 P467

58.   L387 over blank

59.   L390 between the two

60.   L392 containing the 1500-bp promoter

61.   L399 vs 2.6.3 vs Fig 4 Three different “p” styles

62.   L472 receptors (there are more than one)

63.   L474 nuclear

64.   L477 move the “and” to the position after response and before cell.

65.   L489 please insert a space after cancer and also L494

66.   L499 spelling of MbovP467

67.   L505 should this be nucleomodulin singular, rather than plural?

68.   L522 would likely contribute

69.   L524 space after analysis.

70.   L525 supervised RA

Comments on the Quality of English Language

Please see examples included above. 

Author Response

Response to Reviewer 2 Comments

Comments 1

 It would be useful to include the currently proposed name for the taxon in Keywords, the title, the abstract and the main text. Mycoplasmopsis bovis

Response 1

The name “Mycoplasmopsis bovis” has been added throughout in the manuscript.

Comment 2

 References L47 and L50 and L53-57

Response 2

The references have been added (in the introduction in L49, 50, 54 in the revised manuscript).

Comment 3

 It would be useful to mention their obligate lifestyle as host associated

Response 3

The information of obligate lifestyle as host associated for Mycoplasmopsis bovis has been added in introduction (L50 in the revised manuscript).

Comment 4

 L74 As written, this suggests that other species have a SAP11. Is this the intention?

Response 4

The information about the SAP11 protein of Mollicutes has been re-written (L87-88 in the revised manuscript).

Comment 5

Other publications have identified NLS in M. hominis and I believe a nuclease for M. genitalium or M. fermentans. Other examples in the Mycoplasma should be introduced or discussed.

Response 5

This information has been added (L73-80 in the revised manuscript).

Comment 6

 It would be useful to give a unique protein identifier so that P467 can be retrieved from GenBank. Is it conserved in all Mb genomes? Other species? Domains or functional regions other than NLS and LP signal?

Response 6

MbovP467 unique protein identifier is “WP_013954821.1”. This information has been added to the revised manuscript (L129).  The amino acid sequence of MbovP467 is input from NCBI for blast alignment, which is conserved in the genome of Mycoplasma bovis. DUF285 domain is an unknown functional protein in mycoplasma, which is highly conserved only in mycoplasma. This region appears distantly related to leucine-rich repeats. Of course, there are other domains, you can look at the picture below.

Comment 7

L95 is this mutant by transposon or homologous recombination? Is the gene in an operon (in which case there might be polar effects that should be mentioned). Complementation would also be needed.

Response 7

Thank you for pointing this out. T5.141 mutant strains by transposon. Transposon is inserted into the gene body region of Mbov_0467.  The mutated site was at nucleotide (nt) 334bp of Mbov_0467 coding sequence (CDS) or nt 543,809 of the M. bovis HB0801 genome. Please see the image below. This information has also been added to the manuscript (L114-115). The MbovP467 gene is not in the operon. We agree with your comment regarding the complementary strain for mutant T5.141 to compare the results of western blotting, apoptosis, and cell viability between wild M. bovis, mutant and complementary strains. However, this is our initial study to find the role of potential novel nucleomodulin of M. bovis in the pathogenesis of M. bovis as regarding nucleomodulin of M. bovis a limited study is available. So, in the first stage, we compare the results of wild- M. bovis, MbovP467 knock-out mutant T5.141 and control to see the role of novel nucleomodulin MbovP467 in the pathogenesis of M. bovis e.g, apoptosis, cell viability and for the transcriptomic analysis we used recombinant GFP plasmid containing MbovP467 gene. However, our lab is further working on this protein and we acknowledge your suggestion and will make a complementary strain for T5.141 for deeper study to understand the mechanism of MbovP467 protein in M. bovis pathogenesis.

Comment 8

 L108 Was the lipoprotein gene constructed with or without LP signal?

Response 8

We predicted that the MbovP467 protein contained Signal peptide (Sec/SPI) but not Lipoprotein signal peptide (Sec/SPII) using the signal 5.0 online website. The Mbov_0467 gene we constructed contains Sec/SPI signal peptide.

Comment 9

A figure showing the approx. location 0of NLS and the match to consensus would be helpful

Response 9

Can’t understand. Do reviewers want to know the position of NLS in the entire sequence?

If yes please see the location of NLS for MbovP467.

Comment 12

 Section 3.1. How does EGFP enter without NLS? This should be clarified in text. Also, this is a caveat that you cannot tell whether there is additional localization of recombinant dual protein following EGFP dependent nuclear entry.

Response 12

Although EGFP does not have NLS, the entry of EGFP into the nucleus without a nuclear localization signal (NLS) can occur through passive diffusion. EGFP is a relatively small protein (~27 kDa), and it has been observed to enter the nucleus due to its small size and the dynamic nature of nuclear pore complexes. The MbovP467 protein itself is 77Kda and can reach 104Kda after fusion with EGFP (high molecular weight). Therefore, MbovP467 entering the nucleus needs its own NLS, rather than relying on EGFP to carry it.

However, your point is valid and we agree that it should be clarified that while EGFP can enter the nucleus without a typical NLS, the observed nuclear localization of the Mbovp467-EGFP fusion protein via confocal microscopy does not solely confirm the nuclear localization of the Mbovp467 portion. So, in the next step, we did western blotting to confirm the nuclear entry of Mbovp467 using antisera directed against Mbovp467.

Comment 13

How does rMbov467 protein get taken up?

Response 13

The mechanism needs to be explored, our lab is further working on this protein, However, we agree with you that the first complete mechanism of nuclear entry of MbovP467 should be addressed before the publication as other researchers do. Therefore, at this step, we are not presenting this result. We will publish it after finding the mechanism.

Comment 14

Fig 4 shows genes involved in Legionella (I do not think bovines get this), and pertussis (another human disease) and Leishmania (do cattle get this)? The authors have produced a Figure of what the algorithms produce, but context and curation are needed in presentation of results.

Response 14

We appreciate your attention to detail therefore, we added more detail (“The GO enrichment analysis reveals an enrichment of genes associated with Legionella, Leishmania, and pertussis. It's important to note that these pathogens are typically not associated with cattle. The presence of these genes in the results stems from the algorithms used in the analysis, which include a broad range of gene annotations”) in our results to address these points (L435-438 in revised manuscript).

Comment 15

L388 and L410. This is possible, bit overstated here. They have the potential to be involved in infection and pathogenesis in the animal

Response 15

We acknowledge that the significance might be overstated, therefore, we re-wrote the paragraph (L442-444 in the revised manuscript).

Comment 16

L408 No control protein is provided—also it is better to have EGFP fused to an irrelevant protein.

Response 16

Thank you for your thoughtful review of our experiment. We appreciate your comments and understand your concern regarding the absence of a control protein and the suggestion to use EGFP fused to an irrelevant protein.

While we acknowledge the importance of incorporating control proteins to strengthen experimental design, we believe our current setup provides valuable insights into the impact of MbovP467 on cell viability within the context of M. bovis infection. Our findings reveal significant differences in relative cell viability between wild M. bovis strains HB0801, mutant T5.141 (MbovP467 knock-out mutant), and rpEGFP-MbovP467-infected cells, which are consistent with our research objectives.

Moreover, the use of rpEGFP-MbovP467 serves as an effective tool for investigating the specific role of MbovP467 in cell viability, as evidenced by the observed differences compared to wild-type strains. While the inclusion of a control protein could provide additional context, our experimental design aligns with established methodologies in the field, and the results contribute meaningfully to the current body of knowledge.

We understand that replicating experiments with additional controls can enhance the robustness of findings. However, due to time constraints and the widespread use of similar experimental setups in the research community, we believe our findings are valuable and contribute to the advancement of our understanding of M. bovis pathogenesis.

We appreciate your feedback and assure you that we will consider your suggestions for future experiments.

Comment 17

 Fig 2 should be Fig 6 p14

Response 17

This has been corrected (P15, L477 in the revised manuscript).

Comment 18

The legend to Fig 8 should indicate what is in the control. Is it EGFP?

Response 18

The cells treated with PBS were used as control. This information has been added to main text and Figure 8 legend

Comment 19

The Discussion starts with a very long paragraph of 1 page. I would split into multiple and include appropriate transitions.

Response 19

The paragraph has been split (L512 in the revised manuscript).

Comment 20

 L448 are either P475, P467 MTases? This should be mentioned one way or another.

Response 20

These two proteins P475 and P467 are nucleomodulin of M. bovis. Bioinformatically prediction showed not MTases activity of this protein. Experimentally either these are MTases or not  need to be further studied. Therefore, we have mentioned at the end of the discussion (L 581-586 in the revised manuscript).

Comment 21

It would be informative to mention unambiguously whether the many proteins L469/L470 are all known to exist in Bos taurus.

Response 21

 It has been unambiguously mentioned that NF-κB1, NF-κB2, RelA, RelB, and c-Rel are also exist in Bos taurus like almost every mammal cell (L541).

Language points

Comment 1

Title perhaps use “of Mycoplasma” rather than “for”

Response 1

The word “for” has been replaced with “of”

Comment 2

L21 and L29 Depending on journal style, it may not be necessary to include the abbreviated taxa names in parentheses.

Response 2

Abbreviated taxa names in parentheses have been deleted (L22, L31 in the revised manuscript).

Comment 3

L24 This study does not aim to identify, but characterizes one that is identified.

Response 3

The word “characterizes” has been added (L 25in the revised manuscript)

Comment 4

L22 related (non italics)

Response 4

The word “related” has been de-italics (L24 in the revised manuscript).

Comment 5

L26 with a nuclear

Response 5

Has been corrected (L27 in the revised manuscript).

Comment 6

L34 responses (also L59)

Response 6

The word “response” has been replaced with “responses” (L37, L64 in the revised manuscript).

Comment 7

L35 pathways

Response 7

The word “pathway” has been replaced with “pathways” (L36 in the revised manuscript).

Comment 8

L36 processes

Response 8

The word “process” has been replaced with “processes” (L37 in the revised manuscript).

Comment 9

L67 and L68 have different capitalization for Mycoplasma

Response 9

It has been corrected (L72, L73 in the revised manuscript).

Comment 10

L95 and elsewhere  For companies, please include city and State for USA based companies and city and country for others

Response 10

The relevant information (based companies and city and country for others) has been added throughout in the revised manuscript

Comment 11

L97 gentamicin

Response 11

The word “gentamycin” has been replaced with “gentamicin” (L119 in the revised manuscript).

Comment 12

L97 Escherichia should be in full on first use in main text

Response 12

The full form of “Escherichia coli” has been added (L120 in the revised manuscript).

Comment 13

L108 source or reference for pEGFP

Response 13

The source for pEGFP (Shanghai, China) has been added (L132 in the revised manuscript).

Comment 14

L113 please use comma rather than hyphen

Response 14

The word “hyphen” has been replaced with “comma” (L138 in the revised manuscript).

Comment 15

2.4 and 2.5 (and elsewhere) Please use one style of capitalization for headers

Response 15

It has been corrected (L145, L170 in the revised manuscript).

Comment 16

L125 location and company are not needed as given L109

Response 16

Repeated location and company are deleted

Comment 17

L142 and throughout, wild-type rather than wild

Response 17

The word “wild-type” has been added throughout the manuscript

Comment 18

L152 please add city after Abcam…on second instance only company name is needed

Response 18

The has been added city after Abcam (L186, L188 in the revised manuscript).

Comment 19

L153 sources of secondaries

Response 19

Sources of secondary antibodies has been added (L188 in the revised manuscript).

Comment 20

L162 company location for Agilent

Response 20

The company location for Agilent has been added (L196 in the revised manuscript).

Comment 21

L169 company location for ABI

Response 21

The company location for ABI has been added (L203 in the revised manuscript).

Comment 22

L174 processing

Response 22

The word “process” has been replaced with “processing” (L208 in the revised manuscript).

Comment 23

L179, L219, L221, L225 sources or references of databases and tools should be included

Response 23

The references of databases and tools have been added (L213, L253, L255, L259 in the revised manuscript).

Comment 24

L183 summary (lower case)

Response 24

It has been corrected (L217 in the revised manuscript).

Comment 25

Tables Since Treatment is in full, please use Control in full

Response 25

The full form of the word “control” has been added

Comment 26

L187 These do not match the contractions of Cont in Table, same with Table 2 and L214

Response 26

This information has been corrected and now it is the same for both tables (L230, L231, L247, L248 in the revised manuscript).

Comment 27

L202 were aligned

Response 27

It has been corrected (The word “will be” has been replaced with” were” L236 in the revised manuscript).

Comment 28

L203, L206, L208, L209, L210. Theare are terms and symbols (“+” for example) that do not appear in Table 2. Please check the complete footnotes

Response 28

The irrelevant information has been deleted (L240-246 in the revised manuscript).

Comment 29

L231 responses

Response 29

The word “response” has been replaced with “responses” (L265 in the revised manuscript).

Comment 30

L232 or control pEGFP plasmid.

Response 30

The word “or” has been added (L266 in the revised manuscript).

Comment 31

L234 company location also L238

Response 31

The company location has been added (L 269, L272 in the revised manuscript).

Comment 32

L236 add caspases since these are in Table 3

Response 32

Caspases have been added (L271 in the revised manuscript).

Comment 33

L238 the year is not required here

Response 33

The “year” has been deleted (L273 in the revised manuscript).

Comment 34

L245 on the promoter

Response 34

The word “the” has been added (L280 in the revised manuscript).

Comment 35

L247 1500-bp portion of the TNF

Response 35

The “1500-bp portion” has been added (L282 in the revised manuscript).

Comment 36

L247 source/ref for plasmid, Flowjo L285

Response 36

The source and reference for plasmid and Flowjo has been added (L283, L329 in the revised manuscript).

Comment 37

L248-9 is repetitive partially with earlier text, also L254

Response 37

The repeatability has been deleted and the paragraph has been re-written (L285-297 in the revised manuscript).

Comment 38

L256 a control group or “control groups” Please match singular and plural

Response 38

It has been corrected (L293 in the revised manuscript)

Comment 39

L257 which kit?

Response 39

The source has been provided (L294 in the revised manuscript).

Comment 40

L263 Please define CCK-8 and PBS L267

Response 40

The full form of CCK-8 and PBS has been written (L302, L303 in the revised manuscript).

Comment 41

L298, L205 punctated or punctuated?

Response 41

The word “punctuated “has been replaced with “Punctated” L343, L353 in the revised manuscript).

Comment 42

L309 of Nuclear

Response 42

The word “on” has been replaced with “of” (L357 in the revised manuscript).

Comment 43

L312 data not shown

Response 43

We successfully expressed and isolated the MbovP467 protein. Please see the following supplementary material

Comment 44

L314 a western

Response 44

The paragraph has been re-written (L358-378 in the revised manuscript).

Comment 45

L322 protein treated

Response 45

The paragraph has been re-written (L358-378 in the revised manuscript).

Comment 46

Fig 2 align the hours so on same horizontal line. Please also arrange hours for panel B so that they match panel A. It is confusing to have them in opposite arrangements

Response 46

The figure 2B has been deleted because we still not found the mechanism that how MbovP467 enters into the nucleus and the other reason is that the result of mutant T5.141 appeared two times in the same figure. We also explain in your above comment

Comment 47

Fig 2 cytoplasm does not require italics

Response 47

This part has been deleted

Comment 48

L330 and elsewhere. M bovis should always be italicized (multiple examples in legends)

Response 48

  1. bovis has been italicized throughout the manuscript in the revised version

Comment 49

L331 the mutant is duplicated

Response 49

It has been corrected (L383 in the revised manuscript).

Comment 50

L335 Analysis of

Response 50

It has been corrected (the word “on” has been replaced with “of” L85 in the revised manuscript).)

Comment 51

L337 Treatment (to match elsewhere)

Response 51

The word “experimental” has been replaced with “treatment” (L389 in the revised manuscript).

Comment 52

Fig 3 Control (to match full length “treatment”)

Response 52

It has been corrected

Comment 53

L357 are visually

Response 35

The word “were” has been replaced with “are” (L409 in the revised manuscript).

Comment 54

L359 encompassed those involved in inflammatory

Response 54

The words “those involved in” have been added (L411 in the revised manuscript).

Comment 55

L367 Whate are Brite hierarchies? Please include in text

Response 55

The information has been added. Brite hierarchies (Brooklet information hierartical editor) refers to a classification system used in the KEGG database to organize biological entities such as genes, proteins, and pathways into hierarchical categories based on their functional or structural similarities (L419 in the revised manuscript).

Comment 56

P value (Figure 4 panels—please also see note about styles)

Response 56

It has been corrected

Comment 57

L384 P467

Response 57

It has been corrected and now the word “pro-inflammatory” is same throughout the manuscript (L410, L579, and throughout in the revised manuscript).

Comment 58

L387 over blank

Response 58

It has been corrected (L442-444 in the revised manuscript).

Comment 69

L390 between the two

Response 69

The word “the” has been added It has been corrected (L442-444 in the revised manuscript).

Comment 60

L392 containing the 1500-bp promoter

Response 60

It has been corrected (“the 1500-bp promoter” has been added It has been corrected, L451 in the revised manuscript).

Comment 61

L399 vs 2.6.3 vs Fig 4 Three different “p” styles

Response 61

It has been corrected and now style of “p” is same throughout the manuscript

Comment 62

L472 receptors (there are more than one)

Response 62

The word “receptor” has been replaced with “receptors” (L543 in the revised manuscript).

Comment 63

L474 nuclear

Response 63

The word” Nuclear” has been replaced with “nuclear” (L545 in the revised manuscript).

Comment 64

L477 move the “and” to the position after response and before cell.

Response 64

It has been corrected (L549 in the revised manuscript).

Comment 65

L489 please insert a space after cancer and also L494

Response 65

The space has been added (L562 in the revised manuscript).

Comment 66

L499 spelling of MbovP467

Response 66

The spelling has been corrected (L570 in the revised manuscript).

Comment 67

L505 should this be nucleomodulin singular, rather than plural?

Response 67

The word “nucleomodulins” has been replaced with “nucleomodulin” (L577 in the revised manuscript).

Comment 68

L522 would likely contribute

Response 68

The word “likely” has been added (L593 in the revised manuscript).

Comment 69

L524 space after analysis.

Response 69

Space has been added (L595 in the revised manuscript).

Comment 70

L525 supervised RA

Response 70

The word “the RA” has been replaced with “supervised RA” (L596 in the revised manuscript).

Reviewer 3 Report

Comments and Suggestions for Authors

The article is original and very interesting.

The authors aimed to identify a novel nucleomodulin as a virulence-related factor of Mycoplasma bovis.

The topic is very relevant in experimental study of infectious diseases pathogenesis and prevention.

The methodology is ultramodern. Using bioinformatic tools, authors initially predicted MbovP467 as a secreted lipoprotein with nuclear localization signal based on SignalP scores and cNLS Mapper respectively They have demonstrated that that MbovP467 reduces BoMac cell viability and induces the mRNA expression of IL-1β, IL-6, IL-8, TNF-α and apoptosis in BoMac cells.

The conclusions are consistent with the evidence and arguments presented and authors have accurately identified the limitations of the study.

The references are appropriate, including some very relevant author’s previous experience in the field.

I suggest some minor corrections.

In References section authors should follow more carefully The Authors Guide for MDPI article and may add the doi of the articles

Author Response

Response to Reviewer 3 Comments

Comment

In References section authors should follow more carefully The Authors Guide for MDPI article and may add the doi of the articles

Response

Thank you for pointing this out. We agree and appreciate your comment. Therefore, we have checked/updated the references according to the authors' guide for MDPI article and added the DOI of articles.

Round 2

Reviewer 2 Report

Comments and Suggestions for Authors

The revised manuscript is much improved, although a number of points were only addressed in the rebuttal, but should be incorporated into the main document. The manuscript requires further language editing also.

1.       L24 This requires rewording. Perhaps use “characterize a novel …”

2.       L48 pathogen

3.       L50 alike [2].

4.       L52, L67 Some references are bolded, others are not.

5.       L52, L53 [] should be deleted

6.       L 52, 53 Some references are flanked by (), others by []

7.       L69-L76 This information does not fit here as these are not modulins per se

8.       L73 virulence

9.       L73 subsp should not be italicized

10.   L77 needs

11.   L83 to interact

12.   L84 destabilize

13.   L84 of plants

14.   L91 was determined

15.   L100 lung from (remove additional space)

16.   L100 [18] was (no period)

17.   L108 Thermo Fisher

18.   L110 supplemented with 10%

19.   L112 The address is not needed as given L108—see also note 17

20.   L119 which company?

21.   L131 Please delete one of the periods after ml

22.   L137 Please define OD on first use and provide wavelength

23.   L150 should be a new section

24.   L154 Please define MOI if first use

25.   L158 Massachusetts, USA)

26.   L159 and L161 Please delete Massachusetts as already given L158

27.   L179 what is the source/reference for trinity?

28.   L224 Expression of Pro-inflammatory

29.   L234-5. Please place reference L234 and start a new section with “The primer”

30.   L236 List of primers

31.   L247 were treated with

32.   L249 see L108 and delete USA

33.   L251, L261 and throughout. Please use one style of capitalization for headers

34.   L256 After this, 10 uL

35.   L257 All previous mentions of hour have used the units in full. Here it is abbreviated and a space inserted, Fig 2 lacks spaces

36.   L265 Has PBS been defined on first use?

37.   L266 vs L399 and L401/2 caspase or Caspase?

38.   L271 New Jersey (to match style used for states in previous instances), also L276 California

39.   L274 using the Student’s

40.   L277 This section should start with more information about P467. Not a lipoprotein, but has predicted signal sequence, show location of NLS (and SS) within ORF, show location of PARCEL/DUF285 domains, and show comparison of these domains between P467 and P475. It might be useful to include this DUF286-related reference PMID: 20626840. Please also include that it is a monocistronic gene.

41.   L282 purposes

42.   Section 3.1 The authors should include the caveat about EGFP internalization and provide a reference for its proposed mechanism of entry (the information was in the rebuttal letter but was not included in the manuscript)

43.   L297 T5.141 for (insert a space)

44.   L300 Please delete comma after P467

45.   L300 was only

46.   L305 Please remove Panel “A” label

47.   L309 Western blotting

48.   L315 of Differentially Expressed Genes of

49.   L337 up-regulated…..down-regulated (to match L321 style)

50.   L346 reference of the abbreviation BRITE?

51.   L351 and TNF signaling …

52.   Figure 4 panels: pvalue should be two words with italics for p

53.   L363 bacterial names should be italicized

54.   L363 It is

55.   L369 levels….were significantly

56.   L371 pathogenesis.

57.   L72 the effect…on the promoter

58.   Fig 5 title capitalization

59.   L382 Delete “The”

60.   L382 Reduces BoMac..

61.   L388-389 Only one mention of Fig 6 is needed

62.   L412 Expression

63.   Figure 8 requires a title

64.   L417 bacterial name should be italicized

65.   L417 strain

66.   L419-L4290 Company locations have already been given in materials and methods

67.   L422-L475 Is one extremely long paragraph. Perhaps split at L446 with appropriate transition sentence inserted.

68.   L444 is a well

69.   L451 animal names should be italicized

70.   L479 [49

71.   L480 delete a P from PP467

72.   L487 attenuates

73.   The discussion should include comments on conservation of P467 (is it present only in HB801 or is it widely distributed in M. bovis), and mention the responses to points 12, 13, 16 of original review.

Comments on the Quality of English Language

Detailed comments for improvement have been provided. 

Author Response

L24 This requires rewording. Perhaps use “characterize a novel …”

Response

This has been corrected (L24)

L48 pathogen

Response

The word parasite has been replaced with pathogen (L49)

  1. L50 alike [2].

Response

It has been corrected (L50)

  1. L52, L67 Some references are bolded, others are not.

Response

References have been corrected with the same format

  1. L52, L53 [] should be deleted

Response

[] has been deleted (L52, L53)

  1. L 52, 53 Some references are flanked by (), others by []

Response

This has been corrected (L52, L53).

  1. L69-L76 This information does not fit here as these are not modulins per se

Response

This information has been moved to the below paragraph (L81-90).

  1. L73 virulence

Response

It has been corrected (L86)

  1. L73 subsp should not be italicized

Response

Subsp has been de-italicized (86)

  1. L77 needs

Response

It has been corrected (89)

  1. L83 to interact

Response

It has been corrected (L75)

  1. L84 destabilize

Response

The word “destabilizes” has been replaced with “destabilize” (L75)

  1. L84 of plants

Response

The word “plant” has been replaced with “plants” (L76)

  1. L91 was determined

Response

The word “is” has been replaced with “was” (L91)

  1. L100 lung from (remove additional space)

Response

The additional space has been removed (L101)

  1. L100 [18] was (no period)

Response

It has been corrected (L101)

  1. L108 Thermo Fisher

Response

It has been corrected (L109)

  1. L110 supplemented with 10%

Response

It has been corrected (L111)

  1. L112 The address is not needed as given L108—see also note 17

Response

The address has been deleted (L112)

  1. L119 which company?

Response

The information has been updated (L119). (it is from Novagen, Darmstadt, Germany, before mistakenly wrote shanghai, Beijing)

  1. L131 Please delete one of the periods after ml

Reference

It has been deleted (L132)

  1. L137 Please define OD on first use and provide wavelength

Response

 The information has been updated (L139)

  1. L150 should be a new section

Response

It has been corrected (L153)

  1. L154 Please define MOI if first use

Response

The full form of MOI has been written (L156)

  1. L158 Massachusetts, USA)

Response

The word “USA” has been added(161)

  1. L159 and L161 Please delete Massachusetts as already given L158

Response

The word “Massachusetts” (L161, 163)

  1. L179 what is the source/reference for trinity?

response

The reference has been added (L181)

  1. L224 Expression of Pro-inflammatory

Response

It has been corrected (L226)

  1. L234-5. Please place reference L234 and start a new section with “The primer”

Response

It has been corrected (L236)

  1. L236 List of primers

Response

It has been corrected (L238)

  1. L247 were treated with

Response

The word “with” has been added (L249)

  1. L249 see L108 and delete USA

Response

The word “USA” has been deleted (L252)

  1. L251, L261 and throughout. Please use one style of capitalization for headers

Response

253, L263, and throughout. One style of capitalization has been used for headers

  1. L256 After this, 10 uL

Response

“the” has been deleted (L258)

  1. L257 All previous mentions of hour have used the units in full. Here it is abbreviated and a space inserted, Fig 2 lacks spaces

Response

Now it is same throughout the manuscript

  1. L265 Has PBS been defined on first use?

Response

PBS been defined on first use (L257-258)

  1. L266 vs L399 and L401/2 caspase or Caspase?

Response

Now it is same (Caspase) throughout the manuscript)

  1. L271 New Jersey (to match style used for states in previous instances), also L276 California

Response

New Jersey and California have been added (L273, L278)

  1. L274 using the Student’s

Response

The Student’s  has been added (L276)

  1. L277 This section should start with more information about P467. Not a lipoprotein, but has predicted signal sequence, show location of NLS (and SS) within ORF, show location of PARCEL/DUF285 domains, and show comparison of these domains between P467 and P475. It might be useful to include this DUF286-related reference PMID: 20626840. Please also include that it is a monocistronic gene.

Response

The information has been added (L280-288)

  1. L282 purposes

Response

The word “purpose” has been replaced with “purposes” (L293)

  1. Section 3.1 The authors should include the caveat about EGFP internalization and provide a reference for its proposed mechanism of entry (the information was in the rebuttal letter but was not included in the manuscript)

Response

The suggested information has been added in the discussion part (L481-488)

  1. L297 T5.141 for (insert a space)

Response

The space has been added (L335)

  1. L300 Please delete comma after P467

Response

The comma has been deleted (L338)

  1. L300 was only

Response

The space between “was” and “only” has been added (L338)

  1. L305 Please remove Panel “A” label

Response

Panel “A” label gas been removed

  1. L309 Western blotting

Response

It has been corrected (L345)

  1. L315 of Differentially Expressed Genes of

Response

“Differentially Expressed Genes of” has been added (L351)

  1. L337 up-regulated…..down-regulated (to match L321 style)

Response

This has been corrected (L373)

  1. L346 reference of the abbreviation BRITE?

Response

The reference has been added (L383)

  1. L351 and TNF signaling …

Response

“and” has been added (L389)

  1. Figure 4 panels: pvalue should be two words with italics for p

Response

We have used the same style of p value. We also uploaded a new figures file, but it is not updated in the manuscript. We will contact with handling editor to solve this issue

  1. L363 bacterial names should be italicized

Response

Bacterial names have been italicized (L399-400)

  1. L363 It is

Response

The word “Its” has been replaced with “It is” (L400)

  1. L369 levels….were significantly

Response

“levels….were” has been added (L406)

  1. L371 pathogenesis.

Response

This has been corrected (L408)

  1. L72 the effect…on the promoter

Response

“the” has been added (L409)

  1. Fig 5 title capitalization

Response

Fig 5 (now 6) title has been capitalized (L417)

  1. L382 Delete “The”

Response

“The” has been deleted (L420)

  1. L382 Reduces BoMac..

Response

The word “the” has been deleted (L420)

  1. L388-389 Only one mention of Fig 6 is needed

Response

Repetition has been deleted (L426)

  1. L412 Expression

Response

The word “Expression” has been added (L450)

  1. Figure 8 requires a title

Response

The title has been added (L454)

  1. L417 bacterial name should be italicized

Response

Bacterial name has been italicized (L457)

  1. L417 strain

Response

The word “strains” has been replaced with “strain” (457)

  1. L419-L4290 Company locations have already been given in materials and methods

Response

Company locations have been deleted (L459)

  1. L422-L475 Is one extremely long paragraph. Perhaps split at L446 with appropriate transition sentence inserted.

Response

This paragraph has been re-written (L537-540)

  1. L444 is a well

Response

“a” has been added (L509)

  1. L451 animal names should be italicized

Response

Animal name gas been italicized (L516)

  1. L479 [49

Response

Comma has been deleted (L547)

  1. L480 delete a P from PP467

Response

P from pp467 has been deleted (L548)

  1. L487 attenuates

Response

The word “attenuate” has been replaced with “attenuates” (L555)

  1. The discussion should include comments on conservation of P467 (is it present only in HB801 or is it widely distributed in M. bovis), and mention the responses to points 12, 13, 16 of original review.

Response

This information has been added in the discussion (L472-500)

Round 3

Reviewer 2 Report

Comments and Suggestions for Authors

The manuscript is much improved, but there remain a few language issues to revise.

Specific points

1.       Page 1, citation of the left should have the new genus name to match the title L2

2.       L38 experiments, investigating cell viability and the inflammatory

3.       L48 this should be “obligate parasites” as there are many non-pathogenic mycoplasma species

4.       L59and nucleomodulins may govern

5.       L83 The genus names can be abbreviated to M.

6.       L86 Please double check all genus names. The mycoides cluster genus has remained “Mycoplasma”—the old genus name Mycoplasma is retained for certain mycoplasmas, but several new genera are also now described.

7.       L84, L89 Please insert spaces before references.

8.       L100 from a diseased

9.       L102 bold font is not needed for company locations

10.   L108 add growth conditions for BL21 here. Also define LB after Bertani, so that the abbreviation does not require defining L139.

11.   L129, L259, Figure panels. Some instances of numbers and units for “h” have a space, others do not.

12.   L139 (OD600) reached 0.6

13.   L141 MbovP467

14.   L148 Were different purified preps used for different mice? The meaning of “of each purified” is unclear

15.   L152 reached peak titer which

16.   L160 Please add Abcam city on first mention

17.   L170 and L172 California (to match style used for other US states)

18.   Section 2.6.2 Please include what bovine reference sequence was used to map the genes?

19.   L188-189. This is unclear—by “rows” do the authors mean “columns”? Do the authors mean one “read” or one “experiment”?

20.   L204, L205The terms  “total mapped” and “multiple mapped” do not appear in Table 2

21.   L231 and L234 the C. Ltd should not follow China, but should be after company name

22.   L240 Please use consistent capitalization in headers  “on the Promoter”

23.   L256 strain (also L266, L423, L425

24.   L266 HB0801 or mutant

25.   L269 Company location is already given so not needed here—also L273

26.   L281 Please define NCBI on first use; also NLS

27.   L283 and L284 predicted

28.   L287 Furthermore, comparison of the DUF285 motifs of….MbovP475 showed

29.   L323 MbovP0145 has not been mentioned before, so it is unclear why it is included here

30.   L325 DUF285 motifs (or perhaps “domains”_

31.   L330 fluorescence, whereas

32.   L331 plasmids.

33.   L336 were used as protein targets for primary antibodies for

34.   L342 These evidences support

35.   L349 T5.141 infected

36.   L350 only exhibited bands in

37.   L357 upregulation   downregulation to match style used L501/2

38.   L374 These DEGs (also use abbreviation L385, L396, L397, L398

39.   L377 pathway and nuclear

40.   L383 Please delete BRITE here

41.   L408 a potential inflammatory role

42.   Figure legends or methods section. Please include the number of biological and technical replicates that were performed

43.   L437 type

44.   L443 Has PI been defined?

45.   L450 and L454 Different styles of capitalization are used for some Figure titles

46.   L455 flow cytometry

47.   L459 Location of BD is already given

48.   L460 “p” should be in italics to match style used elsewhere

49.   L475 These three taxa are still “Mycoplasma” genus. “475 capri should not be capitalized

50.   A reference for the widespread occurrence of DUF285 in mycoplasmas should be included –PMID 20626840 is appropriate.

51.   L478 into BoMac

52.   L481 has a NLS

53.   L482 not have a NLS; the entry

54.   L485 MbovP467—also L486

55.   L491 analysis by cell

56.   L493 wild-type

57.   L494 We have not used…of an irrelevant L497 strain.

58.   L501 240 DEGS (also L503)

59.   L517 in the immune

60.   L519 regulate

61.   L554 Zhao et al. who demonstrated

Comments on the Quality of English Language

The manuscript is much improved, but there remain a few language issues to revise.

Specific points

1.       Page 1, citation of the left should have the new genus name to match the title L2

2.       L38 experiments, investigating cell viability and the inflammatory

3.       L48 this should be “obligate parasites” as there are many non-pathogenic mycoplasma species

4.       L59and nucleomodulins may govern

5.       L83 The genus names can be abbreviated to M.

6.       L86 Please double check all genus names. The mycoides cluster genus has remained “Mycoplasma”—the old genus name Mycoplasma is retained for certain mycoplasmas, but several new genera are also now described.

7.       L84, L89 Please insert spaces before references.

8.       L100 from a diseased

9.       L102 bold font is not needed for company locations

10.   L108 add growth conditions for BL21 here. Also define LB after Bertani, so that the abbreviation does not require defining L139.

11.   L129, L259, Figure panels. Some instances of numbers and units for “h” have a space, others do not.

12.   L139 (OD600) reached 0.6

13.   L141 MbovP467

14.   L148 Were different purified preps used for different mice? The meaning of “of each purified” is unclear

15.   L152 reached peak titer which

16.   L160 Please add Abcam city on first mention

17.   L170 and L172 California (to match style used for other US states)

18.   Section 2.6.2 Please include what bovine reference sequence was used to map the genes?

19.   L188-189. This is unclear—by “rows” do the authors mean “columns”? Do the authors mean one “read” or one “experiment”?

20.   L204, L205The terms  “total mapped” and “multiple mapped” do not appear in Table 2

21.   L231 and L234 the C. Ltd should not follow China, but should be after company name

22.   L240 Please use consistent capitalization in headers  “on the Promoter”

23.   L256 strain (also L266, L423, L425

24.   L266 HB0801 or mutant

25.   L269 Company location is already given so not needed here—also L273

26.   L281 Please define NCBI on first use; also NLS

27.   L283 and L284 predicted

28.   L287 Furthermore, comparison of the DUF285 motifs of….MbovP475 showed

29.   L323 MbovP0145 has not been mentioned before, so it is unclear why it is included here

30.   L325 DUF285 motifs (or perhaps “domains”_

31.   L330 fluorescence, whereas

32.   L331 plasmids.

33.   L336 were used as protein targets for primary antibodies for

34.   L342 These evidences support

35.   L349 T5.141 infected

36.   L350 only exhibited bands in

37.   L357 upregulation   downregulation to match style used L501/2

38.   L374 These DEGs (also use abbreviation L385, L396, L397, L398

39.   L377 pathway and nuclear

40.   L383 Please delete BRITE here

41.   L408 a potential inflammatory role

42.   Figure legends or methods section. Please include the number of biological and technical replicates that were performed

43.   L437 type

44.   L443 Has PI been defined?

45.   L450 and L454 Different styles of capitalization are used for some Figure titles

46.   L455 flow cytometry

47.   L459 Location of BD is already given

48.   L460 “p” should be in italics to match style used elsewhere

49.   L475 These three taxa are still “Mycoplasma” genus. “475 capri should not be capitalized

50.   A reference for the widespread occurrence of DUF285 in mycoplasmas should be included –PMID 20626840 is appropriate.

51.   L478 into BoMac

52.   L481 has a NLS

53.   L482 not have a NLS; the entry

54.   L485 MbovP467—also L486

55.   L491 analysis by cell

56.   L493 wild-type

57.   L494 We have not used…of an irrelevant L497 strain.

58.   L501 240 DEGS (also L503)

59.   L517 in the immune

60.   L519 regulate

61.   L554 Zhao et al. who demonstrated

Author Response

 Page 1, citation of the left should have the new genus name to match the title L2

We are very sorry we could not understand this comment. What the reviewer wants to say. If the reviewer wants to say that add a new genus name then we have added a new genus name as Mycoplasmopsis bovis. If it is a little change please change it. We will be thankful.

  1. L38 experiments, investigating cell viability and the inflammatory

Response

This information “experiments, investigating cell viability and the inflammatory” has been added (L39)

  1. L48 this should be “obligate parasites” as there are many non-pathogenic mycoplasma species

Response

Before we wrote “obligate parasites” the other reviewer suggested us to write “obligate pathogens”. However, we agree with you it should be “obligate parasites” (L48)

  1. L59and nucleomodulins may govern

Response

The word “protein” has been deleted (L59)

  1. L83 The genus names can be abbreviated to M.

Response

The genus names have been abbreviated to M. (L83, L86)

  1. L86 Please double check all genus names. The mycoides cluster genus has remained “Mycoplasma”—the old genus name Mycoplasma is retained for certain mycoplasmas, but several new genera are also now described.

Response

The genus names have been abbreviated to M. (L86)

  1. L84, L89 Please insert spaces before references.

Response

Spaces before references have been added (L86, L89)

  1. L100 from a diseased

Response

The word “the” has been replaced with “a” (L100)

  1. L102 bold font is not needed for company locations

Response

It has been corrected (L100)

  1. L108 add growth conditions for BL21 here. Also define LB after Bertani, so that the abbreviation does not require defining L139.

Response

The information has been added (L108-110)

  1. L129, L259, Figure panels. Some instances of numbers and units for “h” have a space, others do not.

Response

Now it has been same throughout the manuscript

  1. L139 (OD600) reached 0.6

Response

It has been corrected and the word “was” has been deleted (L140)

  1. L141 MbovP467

Response

You mean we should delete the word “protein”. If yes then the word “protein” has been deleted

  1. L148 Were different purified preps used for different mice? The meaning of “of each purified” is unclear

Response

Now it has been corrected (L149)

  1. L152 reached peak titer which

Response

It has been corrected (L153)

  1. L160 Please add Abcam city on first mention

Response

The city (Massachusetts) has already been added (L162). Please clarify more if you need other information

  1. L170 and L172 California (to match style used for other US states)

Response

It has been corrected (l171, L173)

  1. Section 2.6.2 Please include what bovine reference sequence was used to map the genes?

Response

This information has been added (L185)

  1. L188-189. This is unclear—by “rows” do the authors mean “columns”? Do the authors mean one “read” or one “experiment”?

Response

We are very sorry we cannot understand the question the word “rows” is not mentioned in L188-189

  1. L204, L205The terms  “total mapped” and “multiple mapped” do not appear in Table 2

Response

The information has been updated (L207-208)

  1. L231 and L234 the C. Ltd should not follow China, but should be after company name

Response

This has been corrected (L233, L236)

  1. L240 Please use consistent capitalization in headers  “on the Promoter”

Response

It has been corrected (L242)

  1. L256 strain (also L266, L423, L425

Response

The word “strains” has been replaced with “strain”(L258, L268, L425, L427)

  1. L266 HB0801 or mutant

Response

The word “or” has been added (L268)

  1. L269 Company location is already given so not needed here—also L273

Response

The company location has been deleted (L272). The other reviewer instructed us to write company complete information for FlowJo software. However, we are deleting it. But we do not know if should we delete it or not

  1. L281 Please define NCBI on first use; also NLS

Response

The full form of NCBI (L283) and NLS (L27) have been added

  1. L283 and L284 predicted

Response

The word “predicted” (L286, L287)

  1. L287 Furthermore, comparison of the DUF285 motifs of….MbovP475 showed

Response

The information has been updated (L290-291).

  1. L323 MbovP0145 has not been mentioned before, so it is unclear why it is included here

Response

It is MbovP475 a nucleomodulin of M.bovis strain HB0801. The information has been updated (L291)

  1. L325 DUF285 motifs (or perhaps “domains”

Response

The word “motif” has been replaced with “domains” (L328)

  1. L330 fluorescence, whereas

Response

The “comma” has been added (333)

  1. L331 plasmids.

Response

The full stop has been added (L334)

  1. L336 were used as protein targets for primary antibodies for

Response

The information has been updated (L339)

  1. L342 These evidences support

Response

The word “evidence” has been replaced with “evidences” and “supports” with “support” (L345)

  1. L349 T5.141 infected

Response

The space between T5.141 and infected has been added (352)

  1. L350 only exhibited bands in

Response

It has been corrected (l352, L353)

  1. L357 upregulation   downregulation to match style used L501/2

Response

It has been corrected throughout the manuscript

  1. L374 These DEGs (also use abbreviation L385, L396, L397, L398

Response

The information has been updated (L388, L398, L399, L400, L401) and throughout the manuscript.

  1. L377 pathway and nuclear

Response

The word “and” has been added (L380)

  1. L383 Please delete BRITE here

Response

The word “BRITE” has been deleted (L386)

  1. L408 a potential inflammatory role

Response

The information has been updated (L411)

  1. Figure legends or methods section. Please include the number of biological and technical replicates that were performed

Response

The information has been (L170, L254, L263, L275).

  1. L437 type

Response

It has been corrected (L441)

  1. L443 Has PI been defined?

Response

The full form of PI has been added (L448)

  1. L450 and L454 Different styles of capitalization are used for some Figure titles

Response

It has been corrected throughout the manuscript

  1. L455 flow cytometry

Response

The spellings of flow cytometry are correct. What reviewer want to say?

  1. L459 Location of BD is already given

Response

The company location is deleted (463)

  1. L460 “p” should be in italics to match style used elsewhere

Response

It has been corrected (L464) and now it is same throughout the manuscript

  1. L475 These three taxa are still “Mycoplasma” genus. “475 capri should not be capitalized

Response

It has been corrected (L479)

  1. A reference for the widespread occurrence of DUF285 in mycoplasmas should be included –PMID 20626840 is appropriate.

Response

Please let us know in which section reference should be added

  1. L478 into BoMac

Response

The “into” has been added (L482)

  1. L481 has a NLS

Response

  1. L482 not have a NLS; the entry

Response

The word “a” has been added (L485)

  1. L485 MbovP467—also L486

Response

It has been corrected (L488, L489)

  1. L491 analysis by cell

Response

The word “of” has been replaced with “by” (L494)

  1. L493 wild-type

Response

Wild-type has been added (L496)

  1. L494 We have not used…of an irrelevant L497 strain.

Response

It has been corrected (L497, L500)

  1. L501 240 DEGS (also L503)

Response

The abbreviation “DEGs” has been added (L504, L506)

  1. L517 in the immune

Response

The word “the” has been added (L520)

  1. L519 regulate

Response

The word “regulates” has been replaced with “regulate) (L522)

  1. L554 Zhao et al. who demonstrated

 Response

The comma has been deleted (L558)

Comments on the Quality of English Language

The manuscript is much improved, but there remain a few language issues to revise.

Specific points

  1. Page 1, citation of the left should have the new genus name to match the title L2

We are very sorry we could not understand this comment. What the reviewer wants to say. If the reviewer wants to say that add a new genus name then we have added a new genus name as Mycoplasmopsis bovis. If it is a little change please change it. We will be thankful.

  1. L38 experiments, investigating cell viability and the inflammatory

Response

This information “experiments, investigating cell viability and the inflammatory” has been added (L39)

  1. L48 this should be “obligate parasites” as there are many non-pathogenic mycoplasma species

Response

Before we wrote “obligate parasites” the other reviewer suggested us to write “obligate pathogens”. However, we agree with you it should be “obligate parasites” (L48)

  1. L59and nucleomodulins may govern

Response

The word “protein” has been deleted (L59)

  1. L83 The genus names can be abbreviated to M.

Response

The genus names have been abbreviated to M. (L83, L86)

  1. L86 Please double check all genus names. The mycoides cluster genus has remained “Mycoplasma”—the old genus name Mycoplasma is retained for certain mycoplasmas, but several new genera are also now described.

Response

The genus names have been abbreviated to M. (L86)

  1. L84, L89 Please insert spaces before references.

Response

Spaces before references have been added (L86, L89)

  1. L100 from a diseased

Response

The word “the” has been replaced with “a” (L100)

  1. L102 bold font is not needed for company locations

Response

It has been corrected (L100)

  1. L108 add growth conditions for BL21 here. Also define LB after Bertani, so that the abbreviation does not require defining L139.

Response

The information has been added (L108-110)

  1. L129, L259, Figure panels. Some instances of numbers and units for “h” have a space, others do not.

Response

Now it has been same throughout the manuscript

  1. L139 (OD600) reached 0.6

Response

It has been corrected and the word “was” has been deleted (L140)

  1. L141 MbovP467

Response

You mean we should delete the word “protein”. If yes then the word “protein” has been deleted

  1. L148 Were different purified preps used for different mice? The meaning of “of each purified” is unclear

Response

Now it has been corrected (L149)

  1. L152 reached peak titer which

Response

It has been corrected (L153)

  1. L160 Please add Abcam city on first mention

Response

The city (Massachusetts) has already been added (L162). Please clarify more if you need other information

  1. L170 and L172 California (to match style used for other US states)

Response

It has been corrected (l171, L173)

  1. Section 2.6.2 Please include what bovine reference sequence was used to map the genes?

Response

This information has been added (L185)

  1. L188-189. This is unclear—by “rows” do the authors mean “columns”? Do the authors mean one “read” or one “experiment”?

Response

We are very sorry we cannot understand the question the word “rows” is not mentioned in L188-189

  1. L204, L205The terms  “total mapped” and “multiple mapped” do not appear in Table 2

Response

The information has been updated (L207-208)

  1. L231 and L234 the C. Ltd should not follow China, but should be after company name

Response

This has been corrected (L233, L236)

  1. L240 Please use consistent capitalization in headers  “on the Promoter”

Response

It has been corrected (L242)

  1. L256 strain (also L266, L423, L425

Response

The word “strains” has been replaced with “strain”(L258, L268, L425, L427)

  1. L266 HB0801 or mutant

Response

The word “or” has been added (L268)

  1. L269 Company location is already given so not needed here—also L273

Response

The company location has been deleted (L272). The other reviewer instructed us to write company complete information for FlowJo software. However, we are deleting it. But we do not know if should we delete it or not

  1. L281 Please define NCBI on first use; also NLS

Response

The full form of NCBI (L283) and NLS (L27) have been added

  1. L283 and L284 predicted

Response

The word “predicted” (L286, L287)

  1. L287 Furthermore, comparison of the DUF285 motifs of….MbovP475 showed

Response

The information has been updated (L290-291).

  1. L323 MbovP0145 has not been mentioned before, so it is unclear why it is included here

Response

It is MbovP475 a nucleomodulin of M.bovis strain HB0801. The information has been updated (L291)

  1. L325 DUF285 motifs (or perhaps “domains”

Response

The word “motif” has been replaced with “domains” (L328)

  1. L330 fluorescence, whereas

Response

The “comma” has been added (333)

  1. L331 plasmids.

Response

The full stop has been added (L334)

  1. L336 were used as protein targets for primary antibodies for

Response

The information has been updated (L339)

  1. L342 These evidences support

Response

The word “evidence” has been replaced with “evidences” and “supports” with “support” (L345)

  1. L349 T5.141 infected

Response

The space between T5.141 and infected has been added (352)

  1. L350 only exhibited bands in

Response

It has been corrected (l352, L353)

  1. L357 upregulation   downregulation to match style used L501/2

Response

It has been corrected throughout the manuscript

  1. L374 These DEGs (also use abbreviation L385, L396, L397, L398

Response

The information has been updated (L388, L398, L399, L400, L401) and throughout the manuscript.

  1. L377 pathway and nuclear

Response

The word “and” has been added (L380)

  1. L383 Please delete BRITE here

Response

The word “BRITE” has been deleted (L386)

  1. L408 a potential inflammatory role

Response

The information has been updated (L411)

  1. Figure legends or methods section. Please include the number of biological and technical replicates that were performed

Response

The information has been (L170, L254, L263, L275).

  1. L437 type

Response

It has been corrected (L441)

  1. L443 Has PI been defined?

Response

The full form of PI has been added (L448)

  1. L450 and L454 Different styles of capitalization are used for some Figure titles

Response

It has been corrected throughout the manuscript

  1. L455 flow cytometry

Response

The spellings of flow cytometry are correct. What reviewer want to say?

  1. L459 Location of BD is already given

Response

The company location is deleted (463)

  1. L460 “p” should be in italics to match style used elsewhere

Response

It has been corrected (L464) and now it is same throughout the manuscript

  1. L475 These three taxa are still “Mycoplasma” genus. “475 capri should not be capitalized

Response

It has been corrected (L479)

  1. A reference for the widespread occurrence of DUF285 in mycoplasmas should be included –PMID 20626840 is appropriate.

Response

Please let us know in which section reference should be added

  1. L478 into BoMac

Response

The “into” has been added (L482)

  1. L481 has a NLS

Response

  1. L482 not have a NLS; the entry

Response

The word “a” has been added (L485)

  1. L485 MbovP467—also L486

Response

It has been corrected (L488, L489)

  1. L491 analysis by cell

Response

The word “of” has been replaced with “by” (L494)

  1. L493 wild-type

Response

Wild-type has been added (L496)

  1. L494 We have not used…of an irrelevant L497 strain.

Response

It has been corrected (L497, L500)

  1. L501 240 DEGS (also L503)

Response

The abbreviation “DEGs” has been added (L504, L506)

  1. L517 in the immune

Response

The word “the” has been added (L520)

  1. L519 regulate

Response

The word “regulates” has been replaced with “regulate) (L522)

  1. L554 Zhao et al. who demonstrated

 Response

The comma has been deleted (L558)
